# The whale shark genome reveals patterns of vertebrate gene family evolution

**Milton Tan[1]\*, Anthony K Redmond[2], Helen Dooley[3], Ryo Nozu[4], Keiichi Sato[4,5], Shigehiro Kuraku[6], Sergey Koren[7], Adam M Phillippy[7], Alistair DM Dove[8], Timothy Read[9]**

[1]Illinois Natural History Survey at University of Illinois Urbana-Champaign, Champaign, United States; [2]Smurfit Institute of Genetics, Trinity College Dublin, Dublin, Ireland; [3]University of Maryland School of Medicine, Institute of Marine & Environmental Technology, Baltimore, United States; [4]Okinawa Churashima Research Center, Okinawa Churashima Foundation, Okinawa, Japan; [5]Okinawa Churaumi Aquarium, Motobu, Okinawa, Japan; [6]RIKEN Center for Biosystems Dynamics Research (BDR), RIKEN, Kobe, Japan; [7]National Human Genome Research Institute, National Institutes of Health, Bethesda, United States; [8]Georgia Aquarium, Atlanta, United States; [9]Department of Infectious Diseases, Emory University School of Medicine, Atlanta, United States

**\*For correspondence:**
miltont@illinois.edu

**Abstract** Chondrichthyes (cartilaginous fishes) are fundamental for understanding vertebrate evolution, yet their genomes are understudied. We report long-read sequencing of the whale shark genome to generate the best gapless chondrichthyan genome assembly yet with higher contig contiguity than all other cartilaginous fish genomes, and studied vertebrate genomic evolution of ancestral gene families, immunity, and gigantism. We found a major increase in gene families at the origin of gnathostomes (jawed vertebrates) independent of their genome duplication. We studied vertebrate pathogen recognition receptors (PRRs), which are key in initiating innate immune defense, and found diverse patterns of gene family evolution, demonstrating that adaptive immunity in gnathostomes did not fully displace germline-encoded PRR innovation. We also discovered a new toll-like receptor (TLR29) and three NOD1 copies in the whale shark. We found chondrichthyan and giant vertebrate genomes had decreased substitution rates compared to other vertebrates, but gene family expansion rates varied among vertebrate giants, suggesting substitution and expansion rates of gene families are decoupled in vertebrate genomes. Finally, we found gene families that shifted in expansion rate in vertebrate giants were enriched for human cancer-related genes, consistent with gigantism requiring adaptations to suppress cancer.

## Introduction

Jawed vertebrates (Gnathostomata) comprise two extant major groups, the cartilaginous fishes (Chondrichthyes) and the bony vertebrates (Osteichthyes, including Tetrapoda) (*Venkatesh et al., 2014*). Comparison of genomes between these two groups not only provides insight into early gnathostome evolution and the emergence of various biological features, but also enables inference of ancestral jawed vertebrate traits (*Venkatesh et al., 2014*). The availability of sequence data from many species across vertebrate lineages is key to the success of such studies. Until very recently, genomic data from cartilaginous fishes were significantly underrepresented compared to other vertebrate lineages. The first cartilaginous fish genome, that of *Callorhinchus milii* (known colloquially as ghost shark, elephant shark, or elephant fish), was used to study the early evolution of genes related to bone development and emergence of the adaptive immune system (*Venkatesh et al.,*

*2014*). As a member of the Holocephali (chimaeras, ratfishes), one of the two major groups of cartilaginous fishes, *C. milii* separated from the Elasmobranchii (sharks, rays, and skates) ~420 million years ago, shortly after the divergence from bony vertebrates. Sampling other elasmobranch genomes for comparison is therefore critically important to our understanding of vertebrate genome evolution (*Redmond et al., 2018*).

Until recently, few genetic resources have been available for elasmobranchs in general, and for the whale shark (*Rhincodon typus*) in particular. The first draft elasmobranch genome published was for a male whale shark of Taiwanese origin by *Read et al., 2017*. Famously representing one of Earth's ocean giants, the whale shark is by far the largest of all extant fishes, reaching a maximum confirmed length of nearly 19 m (*McClain et al., 2015*). Due to its phylogenetic position relative to other vertebrates, the scarcity of shark genomes, and its unique biology, the previous whale shark genome assemblies were used to address questions related to vertebrate genome evolution (*Hara et al., 2018*; *Marra et al., 2019*), the relationship of gene evolution in sharks and unique shark traits (*Hara et al., 2018*; *Marra et al., 2019*), as well as the evolution of gigantism (*Weber et al., 2020*). A toll-like receptor (TLR) similar to TLR21 was also found in this first whale shark genome draft assembly, suggesting that TLR21 was derived in the most recent common ancestor (MRCA) of jawed vertebrates. While this represented an important step forward for elasmobranch genomics, the assemblies were fragmentary, and substantial improvements to the genome contiguity and annotation were expected from reassembling the genome using PacBio long-read sequences (*Read et al., 2017*).

Despite the relative lack of genomic information prior, much recent work has focused upon further sequencing, assembling, and analyzing of the whale shark nuclear genome (*Hara et al., 2018*; *Read et al., 2017*; *Weber et al., 2020*). Hara et al. reassembled the published whale shark genome data and sequenced transcriptome data from blood cells sampled from a different individual for annotation (*Hara et al., 2018*). Alongside the work on the whale shark genome, genomes have also been assembled for the bamboo shark (*Chiloscyllium punctatum*), cloudy catshark (*Scyliorhinus torazame*), white shark (*Carcharodon carcharias*), and white-spotted bamboo shark (*Chiloscyllium plagiosum*). Comparative analyses of shark genomes supported numerous evolutionary implications of shark genome evolution, including a slow rate of shark genome evolution, a reduction in olfactory gene diversity, positive selection of wound healing genes, proliferation of CR1-like LINEs within introns related to their larger genomes, and rapid evolution in immune-related genes (*Hara et al., 2018*; *Marra et al., 2019*; *Weber et al., 2020*; *Zhang et al., 2020*).

Long-read sequencing is an important factor in assembling longer contigs to resolve repetitive regions which comprise the majority of vertebrate genomes (*Koren et al., 2017*). Herein, we report on the best gapless assembly of the whale shark genome to date, based on de novo assembly of long reads obtained with the PacBio single molecule real-time sequencing platform. We used this assembly and new annotation in a comparative genomic approach to investigate the origins and losses of gene families, aiming to identify patterns of gene family evolution associated with major early vertebrate evolutionary transitions. Building upon our previous finding of a putative TLR21 in the initial draft whale shark genome assembly, we performed a detailed examination of the evolution of jawed vertebrate pathogen recognition receptors (PRRs), which are innate immune molecules that play a vital role in the detection of pathogens. Despite their clear functional importance, PRRs (and innate immune molecules in general), have been poorly studied in cartilaginous fishes until now. Given that cartilaginous fishes are the most distant evolutionarily lineage relative to humans to possess both an adaptive and innate immune system, the study of their PRR repertoire is important to understanding the integration of the two systems in early jawed vertebrates. For example, previous work has shown several deuterostome invertebrate genomes possess greater expanded PRR repertoires when compared to relatively conserved repertoires found in bony vertebrates, which suggests that adaptive immunity may have negated the need for many new PRRs in jawed vertebrates (*Huang et al., 2008*; *Rast et al., 2006*; *Smith et al., 2013*), although this hypothesis has not been formally tested. Using the new whale shark genome assembly, we therefore investigated the repertoires of three major PRR families: NOD-like receptors (NLRs), RIG-like receptors (RLRs), and TLRs. Next, we compared the rates of functional genomic evolution in multiple independent lineages of vertebrates in which gigantism has evolved, including the whale shark, to test for relationships between gigantism and genomic evolution among vertebrates. Finally, we studied whether gene families that have shifted in gene duplication rates were enriched for orthologs of known cancer-

related genes. Larger-bodied organisms tend to have lower cancer rates than expected given their increased numbers of cells relative to smaller-bodied organisms (*Peto et al., 1975*), suggesting genes involved in cancer suppression may evolve differently in vertebrate giants. Recent research in giant mammals such as elephants and whales supports this hypothesis and has identified selection or duplication of various gene families that are related to suppressing cancer in humans (*Abegglen et al., 2015*; *Sulak et al., 2016*; *Tollis et al., 2019*). Hence, we studied whether gene families that have shifted in gene duplication rates were enriched for orthologs of known cancer-related genes.

## Results and discussion

### Gapless genome assembly

We added to our previously sequenced short-read Illumina data (~30× coverage) by generating 61.8 Gbp of long-read PacBio sequences; relative to a non-sequencing-based estimate of genome size of 3.73 Gbp (*Hara et al., 2018*), this was an expected coverage of long-read sequences of about 16× coverage, for a total of ~46× coverage. The new whale shark genome assembly represented the best gapless assembly to date for the whale shark (*Supplementary file 1*). The total length of the new assembly was 2.96 Gbp. The total size of the assembly was very similar to the genome size estimated from the *k*-mer-based approach GenomeScope of ~2.79 Gbp, suggesting the genome is fairly complete. On the other hand, it was smaller than a non-sequencing-based estimate of the whale shark genome size of 3.73 Gbp by *Hara et al., 2018*, which suggests that sections of the genome, potentially comprising primarily repetitive elements, are still missing. Repetitive elements were annotated to comprise roughly 50.34% of the genome assembly (Appendix 1).

The new assembly had 57,333 contigs with a contig N50 of 144,422 bp, or fewer contigs than the number of scaffolds of previous assemblies, and a higher contig N50, representing a dramatic improvement in gapless contiguity compared to the existing whale shark genome assemblies (*Supplementary file 1*). This higher contiguity at the contig level (vs. scaffold level) was also better than the published *Callorhinchus*, brownbanded bamboo shark, cloudy catshark, and white shark genomes (*Hara et al., 2018*; *Marra et al., 2019*; *Venkatesh et al., 2014*). The scaffolded genome had 39,176 scaffolds and a scaffold N50 of 344,460 bp. Relative to the previously published most contiguous assembly by *Weber et al., 2020*, while they reported a far higher scaffold N50 of ~2.56 Gb (3.13 Gb ≥ 200 bp), our scaffolded assembly had far fewer scaffolds (3.3M sequences, 139,611 sequences ≥200 bp) (*Supplementary file 1*).

Based on GenomeScope, the whale shark genome had an estimated level of heterozygosity of ~0.0797–0.0828%, consistent with the *k*-mer coverage plot showing a unimodal distribution (*Appendix 1—figure 1*). Comparison of the *k*-mer profile with the presences of *k*-mers in the assembly revealed that they were relatively concordant, with no indication that there were many *k*-mers that were represented twice in the assembly (which could be due to a diploid individual having phased haplotypes assembled into separate contigs) (*Appendix 1—figure 2*). Mapping the reads to the genome assembly and calling SNPs using freebayes provided a similar estimate of 2,189,244 SNPs, a rate of 0.0739% heterozygosity, or an average of an SNP every 1353 bases. This suggests that the heterozygosity of the whale shark genome is relatively low.

To assess gene completeness, we first used BUSCO v2 (*Simão et al., 2015*). Of 2586 orthologs conserved among vertebrates searched by BUSCO (*Simão et al., 2015*), we found 2033 complete orthologs in the whale shark genome, of which 1967 were single-copy and 66 had duplicates; 323 orthologs were detected as fragments, while 230 were not detected by BUSCO. With 78.7% complete genes, this represents a marked improvement over the previous whale shark genome assembly, which only had 15% complete BUSCO genes (*Hara et al., 2018*). We also evaluated gene completeness using a rigid one-to-one ortholog core vertebrate gene (CVG) set that is better tuned for finding gene families in elasmobranchs (*Hara et al., 2015*) implemented in gVolante server (*Nishimura et al., 2017*; *Nishimura et al., 2019*), we found that 85% of CVGs were complete and found that 97.4% of CVGs included partial genes, which compares favorably to completeness statistics in other shark assemblies (*Hara et al., 2018*; *Supplementary file 2*). The gene content of this whale shark assembly was thus quite complete and informative for questions regarding vertebrate gene evolution.

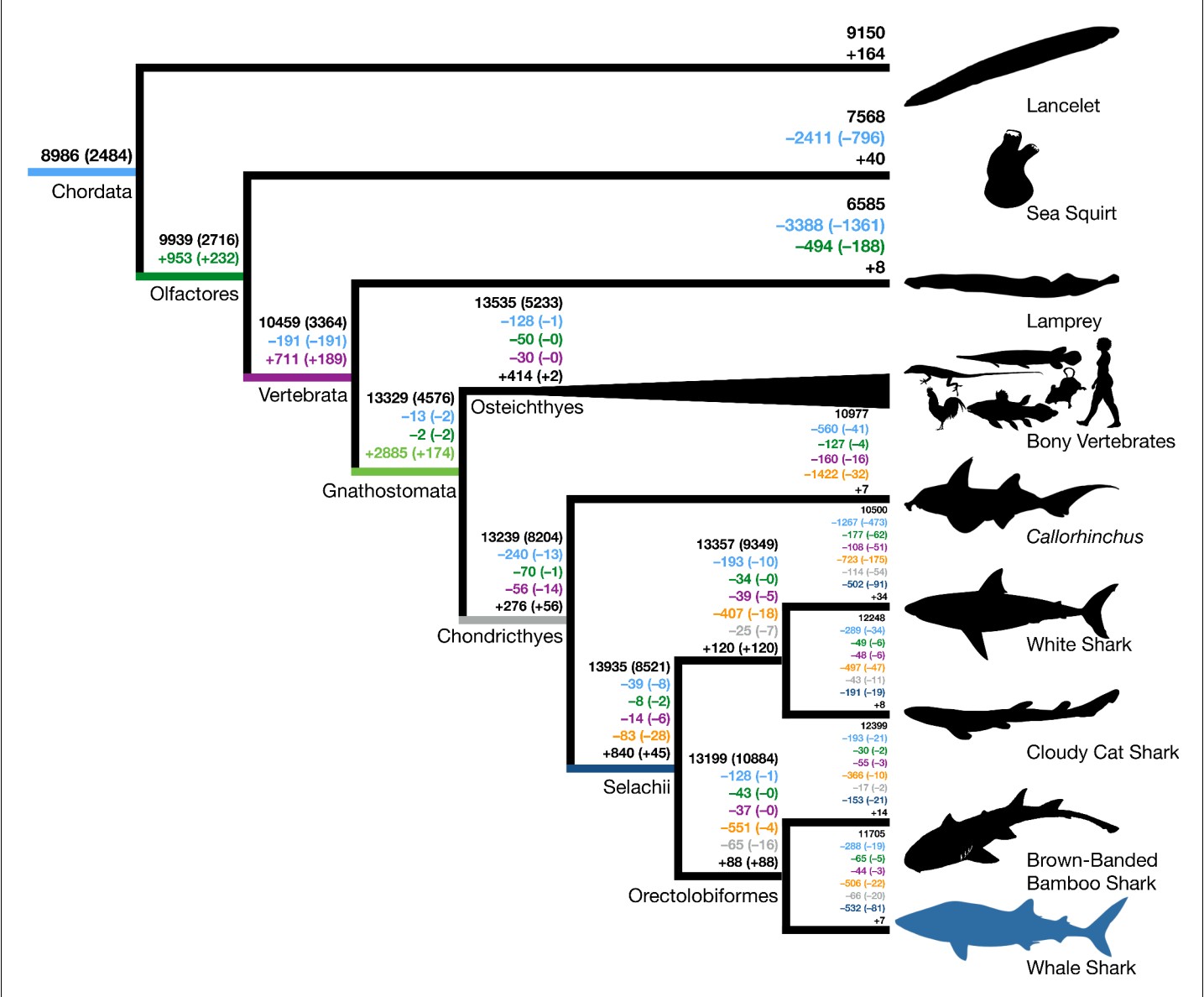

**Figure 1.** Origins and losses of vertebrate gene families. Above the branch in black is the total number of gene families inferred to be present in the most recent common ancestor at that branch; the number in parentheses indicates the number of gene families conserved in all descendants of that branch. Numbers preceded by + and – indicate the number of gene families inferred to be gained or lost along that branch, respectively. Gains and losses are color-coded based on the branch where these gene families originated. Light blue indicates gene families present in the most recent common ancestor of chordates, green indicates gene families that originated in the most recent common ancestor of tunicates and vertebrates (Olfactores), purple indicates vertebrate-derived gene families, orange indicates gnathostome-derived gene families, gray indicates chondrichthyan-derived gene families, while dark blue indicates shark-derived gene families. Negative numbers within parentheses indicate gene family losses that are unique to that branch (as opposed to gene families that were also lost along other branches). Positive colored numbers within parentheses indicate novel gene families conserved in all descendants ('core' gene families).

## Ancestral vertebrate genome evolution

We sought to use the new whale shark genome assembly to infer the evolutionary history of protein-coding gene families (i.e. orthogroups) across vertebrate phylogeny, aiming to provide insight into the evolution of biological innovations during major transitions in vertebrate evolution. Orthogroups are defined as all genes descended from a single gene in the common ancestor of the species considered (*Emms and Kelly, 2015*); hence, they are dependent to a degree on the phylogenetic breadth of species included in the analysis. Genes from the proteomes for 37 chordate species

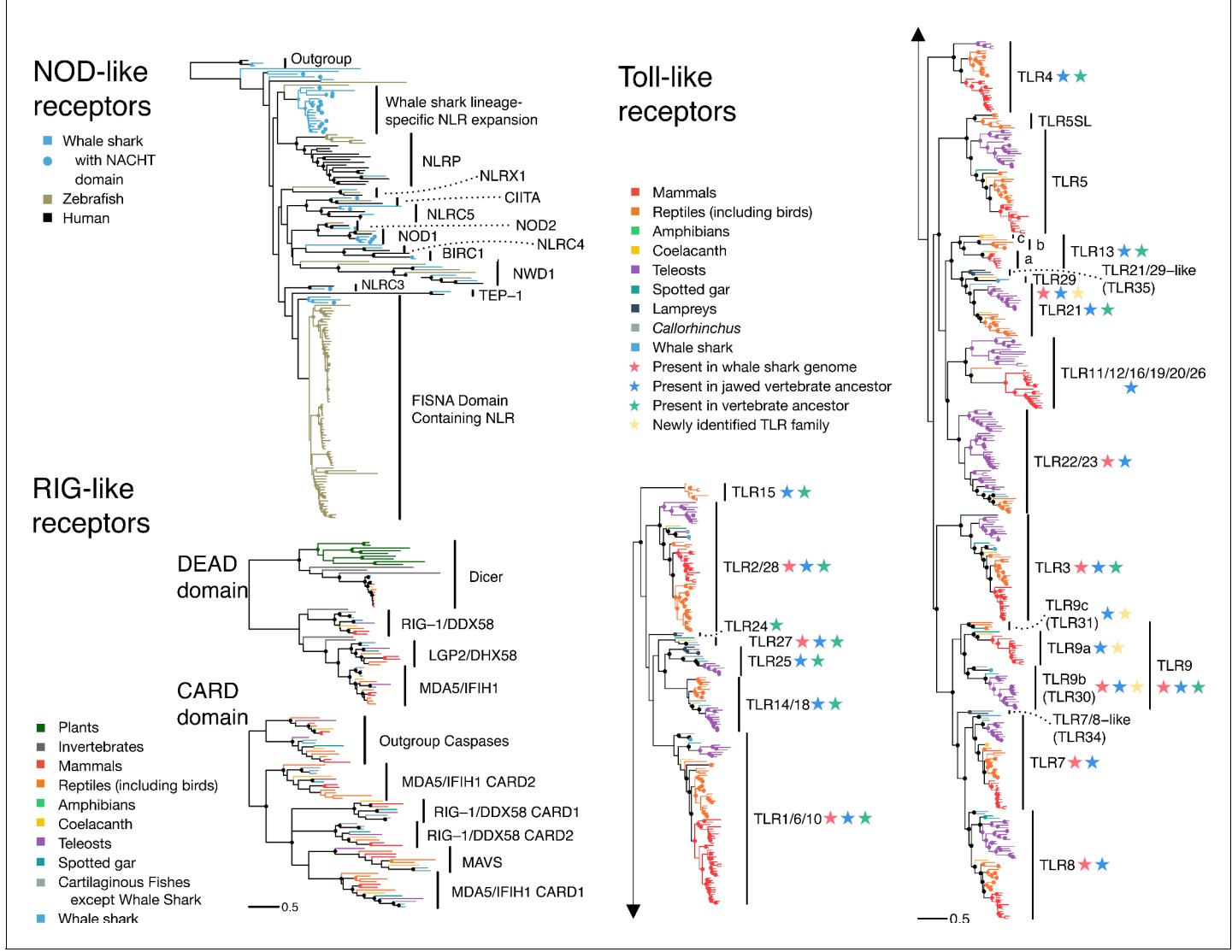

**Figure 2.** The pathogen recognition receptor (PRR) repertoire of whale shark. Nodes supported ≥95% UFBOOT indicated with a dot. For NOD-like receptors, NLRs in whale shark with a NACHT domain are indicated by a dot at the tip. See also *Figure 2—figure supplements 1–2*. For RIG-like receptors (RLRs), branches are colored by gene, except for RLRs in whale shark which are colored distinctly and labeled by a dot at each tip. See also *Figure 2—figure supplement 3*. For toll-like receptors (TLRs), each clade represents a separate TLR except families found within TLR13 are also labeled a (TLR13a), b (TLR32), and c (TLR33). TLR families are also labeled by stars indicating whether they were present in the whale shark genome, present in jawed vertebrate ancestor, present in the vertebrate ancestor, and novel to this study. See also *Figure 2—figure supplement 4*.

The online version of this article includes the following figure supplement(s) for figure 2:

**Figure supplement 1.** Phylogenetic analysis of NOD-like receptors (NLRs) from whale shark, zebrafish, and human.

**Figure supplement 2.** Detailed analysis of NOD1 evolution.

**Figure supplement 3.** Phylogenetic analyses of whale shark and jawed vertebrate RIG-like receptors (RLRs), DICER, and mitochondrial antiviral signaling (MAVS).

**Figure supplement 4.** Phylogenetic tree of vertebrate toll-like receptors (TLRs), including new whale shark sequences.

deriving primarily from Ensembl and RefSeq (*O'Leary et al., 2016*; *Yates et al., 2020*; *Supplementary file 3*), including 35 representative vertebrates, a sea squirt (*Ciona*), and a lancelet (*Branchiostoma*) were assigned to 18,435 orthologous gene families using OrthoFinder, which were assigned to a mean of 10,688 orthogroups per genome (*Supplementary file 4*). We then inferred the history of gene family origin and loss by comparing the presence and absence of gene families across species. Although accurate inference of gene family size evolution in vertebrates may benefit from further taxon sampling among invertebrates, *Branchiostoma* is notable among animals for

retaining a relatively high number of gene families present in the animal stem branch (*Richter et al., 2018*), hence it is one of the best outgroup species to vertebrates to study the origins of novel genes within the vertebrate clade.

We inferred a consistent increase in the total number of gene families from the root to the MRCA of Gnathostomata, but only slight increases following this in the MRCAs of bony fishes and cartilaginous fishes (black numbers, *Figure 1*). We also found numbers of novel gene families increased from the root to a peak in the MRCA of gnathostomes, and then novelty decreased precipitously toward the bony and cartilaginous fish descendants (numbers indicated by + symbol, *Figure 1*). Gene families conserved in all members of a clade may be considered core genes. There was also an increase in the number of core genes of each ancestor from the most inclusive to the least inclusive clades, as expected with decreasing phylogenetic breadth of the less inclusive clades (black parenthetical numbers, *Figure 1*). We found that a decreasing number of novel core gene families in vertebrate ancestors were retained between the MRCA of Olfactores (tunicates + vertebrates) and the MRCA of Gnathostomata (parenthetical numbers indicated by +, *Figure 1*), in contrast to the general pattern of increasing numbers of novel gene families along these same branches. Overall, this implied that the origin of jawed vertebrates established a large proportion of novel gene families of both bony vertebrates and cartilaginous fishes, with variable retention among descendant lineages.

The inclusion of multiple chondrichthyan lineages was important in the inference of gnathostome-derived gene families. The selachians (true sharks) lost fewer gene families than *Callorhinchus* (144 vs. 2269 overall gene families, 83 vs. 1422 gnathostome-derived gene families), but future improved taxon sampling may further increase this estimate. Additional high-quality genomes of holocephalans may demonstrate that some of these losses in *Callorhinchus* are lineage-specific or due to assembly or annotation errors, and the addition of batoids (skates and rays) could recover gene families independently lost in holocephalans and selachians. Thus, increasing chondrichthyan taxon sampling allowed for more confidence in the origin and loss of gene families in vertebrate history and assignment of hundreds of genes as having originated prior to the MRCA of gnathostomes.

The burst in emergence of novel gene families that we observed along the ancestral jawed vertebrate branch coincides with the two rounds (2R) of whole-genome duplication that occurred early in vertebrate evolution, resulting in gene duplicates referred to as ohnologs (*Braasch et al., 2016*; *Ohno et al., 1968*; *Singh et al., 2015*). Hypothetically, divergent ohnologs may be erroneously assigned to novel gene families and artifactually inflate our estimate for gene family birth along the ancestral jawed vertebrate branch. To estimate the potential extent of ohnolog family splitting, we compared the 2885 gene families inferred as novel at the base of jawed vertebrates to ohnolog families previously inferred by *Singh et al., 2015*. Generally, most ohnologs had all their copies assigned to single gene families (1131–1609 ohnologs per species). We found that only 157 (5.4% of 2885 gene families) of gene families that we inferred to have originated in the MRCA of jawed vertebrates corresponded to split ohnologs. Hence, the split ohnologs are not a large proportion of the novel gene families in the jawed vertebrate ancestor. In addition, we also found that only three gene families were inferred to be derived at the MRCA of teleosts, coinciding with the teleost-specific genome duplication, which also supports that gene family inference is robust to genome duplication. This finding reinforces the importance of this evolutionary transition for genomic novelty, not just due to the vertebrate 2R whole-genome duplication, but also through the addition of novel gene families.

Next, we tested whether gene families that were gained or lost during vertebrate evolution were enriched for certain GO (gene ontology) or Pfam annotations (*Supplementary file 5*), potentially indicating functional genomic shifts preceding the origin of these clades. Functional annotations were annotated using InterProScan 5.32–71.0 and Kinfin 1.0 (*Laetsch and Blaxter, 2017*; *Supplementary file 5*). Functional enrichment was determined using the Fisher's exact test (tests summarized in *Supplementary file 6*). Overall, there were 8700 gene families (47.2%) annotated for GO functional terms and 14,727 gene families (79.9%) annotated for Pfam protein domains. For example, for the 711 novel gene families in the MRCA of Olfactores, we found an enrichment of connexin function (*Supplementary file 7*). This is consistent with prior work that determined that the origin of connexin gap junction proteins among chordates was in the MRCA in Olfactores (*Alexopoulos et al., 2004*). We also found enrichment of ankyrin repeat domains, a motif found widely across eukaryotes which has diverse functions in mediation of protein–protein interactions (*Li et al., 2006*), and hence may be involved in the evolution of novel protein complexes in Olfactores. Also, specific to the evolution of the whale shark, we inferred seven novel gene families and a

loss of 1501 gene families. Neither of these sets of gene families were enriched for any functional terms, suggesting whale shark-specific traits are not attributed to functional genomic shifts due to the origins or losses of gene families.

Among the novel proteins in the MRCA of vertebrates, we found enrichment of several protein domain types including rhodopsin family 7-transmembrane (7-TM) receptor domains, immunoglobulin V-set domain, collagen triple helix repeats, zona pellucida domain, and C2H2-type zinc finger domain (*Supplementary file 8*). The enrichment of collagen function is consistent with the importance of these collagens at the origin of vertebrates and their potential involvement in origin of vertebrate traits, such as bone and teeth (*Boot-Handford and Tuckwell, 2003*). The enrichment of the zona pellucida domain at the origin of vertebrates is consistent with previous evidence showing that zona pellucida proteins likely originated in vertebrates (*Litscher and Wassarman, 2014*). Inner-ear proteins also contain the zona pellucida domain, making its appearance in the vertebrate ancestor coincident with the origin of inner ears (*Popper et al., 1992*). The 7-TM domain proteins include a wide variety of receptors but were not enriched for any particular GO term. Some example receptors include those involved in binding a variety of ligands (e.g. fatty acids, neuropeptides, and hormones) and receptors with immune relevance (e.g. chemokine, bradykinin, and protease-activated receptors). The immunoglobulin V-set domain was found in several proteins, most which had roles in cell adhesion and other functions. We also found enrichment among novel vertebrate genes for the C2H2-type zing finger domain, a well-characterized zinc finger domain primarily responsible for nucleotide–protein, as well as protein–protein interactions (*Brayer and Segal, 2008*; *Wolfe et al., 2000*). These novel genes were also not enriched for any particular GO term, but play a role in a variety of developmental signaling pathways and cell cycle regulation (*Supplementary file 8*). The enrichment of these varied functional protein domains in the MRCA of vertebrates demonstrates their importance in the origin of diverse vertebrate traits, including responding to stimuli, fertilization, immunity, and signaling. Although the origins of some of these gene families were in the vertebrate ancestor, subsequent gene diversification in jawed vertebrates continued to increase the functional diversity of these gene families, such as in the collagens which were duplicated in the jawed vertebrate genome duplication (*Haq et al., 2019*; *Wada et al., 2006*).

In the MRCA of jawed vertebrates, we found enrichment of a variety of immune-related protein domains including immunoglobulin V-set domain, immunoglobulin C1-set domain, and interleukin-8-like small cytokines, with functional enrichment of immune response and hormone activity. Immunoglobulin domain containing gene families included many immunoglobulins, interleukins, interleukin receptors, T-cell receptors, sialic acid-binding immunoglobulin-type lectins (Siglec proteins), chemokines, cluster of differentiation (CD) proteins, and MHC proteins (*Supplementary file 9*), consistent with the evolution of immunoglobulin/T-cell receptor-based adaptive immunity in gnathostomes (*Boehm, 2012*; *Kaiser et al., 2004*). We also found enrichment for hormone activity, related to the origin of genes for many hormones at the origin of vertebrates (*Supplementary file 9*). This finding complemented previous work that identified hormones with a role in mammal homeostasis originating in the MRCA of jawed vertebrates, but it emphasizes that hormone activity is also a predominant function among earlier novel vertebrate gene families (*Hara et al., 2018*).

Differences between bony vertebrates and cartilaginous fishes might be due to functional differences in gene families specific to each lineage. For gene families exclusive to bony vertebrates (including the 414 gene families derived in the MRCA of bony vertebrates and 366 gene families lost in cartilaginous fishes), we found several enriched sets of protein domains and functions including GPCR domain, lectin C-type domain, and C2H2-type zinc finger proteins (*Supplementary files 10* and *11*). GPCRs are 7-TM proteins that transmit signals in response to extracellular stimuli to G proteins (*Pierce et al., 2002*). This enrichment of GPCR protein function is consistent with the relative paucity of these receptors in cartilaginous fishes, noted previously (*Marra et al., 2019*). We found many of the GPCRs gained in the MRCA of bony vertebrates were olfactory receptors, which is also consistent with the relatively low number of olfactory receptors noted in cartilaginous fishes previously (*Hara et al., 2018*; *Marra et al., 2019*). We also found that one of the GPCR gene families included MAS1 and its relatives. MAS1 is important in response to angiotensin and regulating blood pressure, and although sharks produce angiotensin I (*Takei et al., 1993*), the precursor to angiotensin-(1–7), the lack of MAS1 and related receptors in cartilaginous fishes suggests that such responses are mediated by alternative receptors and that blood pressure regulation is distinct between cartilaginous fishes and bony vertebrates. Among the lectin C-type domain proteins, we found no

orthologs of the NK gene cluster in cartilaginous fishes (e.g. CD69, KLRC), a conserved complex of genes found across bony vertebrates, which implies potential differences in the natural killer complex in cartilaginous fishes (*Kelley et al., 2005*). Gene families lost in cartilaginous fishes are also enriched for loss of KRAB box domain, which play a role in transcription repression factors (*Margolin et al., 1994*). There was also enrichment for genes including the C-type lectin domain, which bind a variety of ligands and have functions including playing roles in immunity (*Brown et al., 2018*). By contrast, we did not find enrichment in domains or functions among the gene families derived in cartilaginous fishes, which may in part be due to fewer annotations among gene families that are not present in bony vertebrates (Appendix 2). In summary, functional genomic differences between bony vertebrates and cartilaginous fishes are due to differences in the presence of gene families in bony vertebrates, with some related to immunity, chemosensation, and signaling.

Our analyses imply a dynamic history of gene family gain and loss across early vertebrate evolution. Of particular importance was the number of gene families gained in the MRCA of jawed vertebrates in establishing the gene families that are present in bony vertebrates and cartilaginous fishes, with these novel gene families being enriched for immune-related functions. The whale shark genome provided an important additional resource to study the origins of gene families in early vertebrate evolution.

## Evolution of jawed vertebrate innate immune receptors

Cartilaginous fishes are the most distant human relatives to possess an adaptive immune system based on immunoglobulin antibodies and T-cell receptors (*Dooley, 2014*; *Flajnik and Kasahara, 2010*). This has driven extensive functional study of their adaptive immune system and an in-depth, although controversial, analysis of the evolution of adaptive immune genes in the elephant shark genome (*Dooley, 2014*; *Redmond et al., 2018*; *Venkatesh et al., 2014*). By comparison, the cartilaginous fish innate immune system has been overlooked (*Krishnaswamy Gopalan et al., 2014*), despite its importance to understanding the impact that the emergence of adaptive immunity had on innate immune innovation. For example, some previous analyses of deuterostome invertebrate genomes identified greatly expanded PRR repertoires. Yet, vertebrate PRR repertoires are considered to be highly conserved, leading to the suggestion that the need for vast PRR repertoires in vertebrates was superseded by the presence of adaptive immunity in vertebrates (*Huang et al., 2008*; *Rast et al., 2006*; *Smith et al., 2013*). Notable exceptions to this include an expansion of TLRs in codfishes due to a proposed loss of CD4 and MHC class II (*Solbakken et al., 2017*), and expansion of fish-specific NLRs in some other teleosts (*Howe et al., 2016*; *Stein et al., 2007*). As such, we sought to use the whale shark genome to determine whether cartilaginous fishes have a similar PRR set to bony vertebrates, with which they share an adaptive immune system, and also search for evidence of PRR expansions aiming to better understand vertebrate innate immune evolution. To this end, we used BLAST to identify whale shark sequences corresponding to three major PRR families – NLRs, RLRs, and TLRs – and reconstructed their phylogeny among published, curated vertebrate PRR gene datasets.

NLRs are intracellular receptors that detect a wide array of pathogen- (PAMPs) and damage-associated molecular patterns (DAMPs) (e.g. flagellin, extracellular ATP, glucose) (*Fritz and Kufer, 2015*). We identified 43 putative NLRs in the whale shark. We found direct orthologs of almost all human NLRs (UFBOOT=100 for all; *Supplementary file 12*), of which 23 contained a clearly identifiable NACHT domain (a signature of NLRs) (NOD1, NOD2, CIITA, NLRC5, NLRC3) while other putative orthologs did not (NLRX1, NLRC4, BIRC1, NWD1, TEP-1, and NLRP). While inclusion of these sequences lacking an apparent NACHT might seem questionable, the false-negative rate for NACHT domain detection is high, even for some human NLRs. The presence of these orthologs in whale shark indicates the presence of a conserved core NLR repertoire in jawed vertebrates (*Supplementary file 12*, *Figure 2* and *Figure 2—figure supplement 1*, Appendix 3). Surprisingly, we found three orthologs of NOD1 in the whale shark, which is a key receptor for detection of intracellular bacteria, rather than a single copy as in humans (ultrafast bootstrap support, UFBOOT=100; *Figure 2—figure supplement 1* and *Figure 2—figure supplement 2*). Further analyses intimate that the three NOD1 copies resulted from tandem duplication events in the ancestor of cartilaginous fishes (Appendix 3). Sequence characterization suggests that all three of the whale shark NOD1s possess a canonical NACHT domain and so should retain a NOD1-like binding mechanism, but may have unique recognition specificity (*Figure 2—figure supplement 2*,

Appendix 3). Thus, we hypothesize that the three NOD1s present in cartilaginous fishes potentiate broader bacterial recognition or more nuanced responses to intracellular pathogens. In contrast to the scenario observed for NOD1, we did not find NACHT domain containing orthologs of any of the 14 human NLRP genes, many of which activate inflammatory responses (*Schroder and Tschopp, 2010*), in whale shark, and only a single sequence lacking a detectable NACHT domain (*Supplementary file 12*, *Figure 2* and *Figure 2—figure supplement 1*, Appendix 3). However, we did identify an apparently novel jawed vertebrate NLR gene family that appears to be closely related to the NLRPs (UFBOOT=67; *Figure 2* and *Figure 2—figure supplement 1*). This family has undergone significant expansion in the whale shark (UFBOOT=100; *Figure 2* and *Figure 2—figure supplement 1*), and we tentatively suggest that this may compensate for the paucity of true NLRPs in whale shark. Nonetheless, these results imply that the NLR-based inflammasomes in humans and whale sharks are not directly orthologous, and hence that NLR-based induction of inflammation and inflammation-induced programmed cell death (*Schroder and Tschopp, 2010*) are functionally distinct in human and whale shark. Interestingly, each of the vertebrate species we examined (human, zebrafish, and whale shark) has independently expanded different NLR subfamilies relative to the other species included in the analysis, with NLRP genes expanded in human (clade UFBOOT=99; *Figure 2* and *Figure 2—figure supplement 1*) and the previously identified 'fish-specific' FISNA in zebrafish (clade UFBOOT=86; *Figure 2* and *Figure 2—figure supplement 1*). For the latter, we unexpectedly found a whale shark ortholog (UFBOOT=74), suggesting this gene was present in the jawed vertebrate ancestor and is not a teleost novelty. In all, while we found evidence for a core set of NLRs in jawed vertebrates, our analyses also show that multiple, independent NLR repertoire expansions, with probable immunological relevance, have occurred during jawed vertebrate evolution despite the presence of the adaptive immune system.

RLRs are intracellular receptors that detect viral nucleic acid and initiate immune responses through mitochondrial antiviral signaling (MAVS) protein (*Mukherjee et al., 2014*). Bony vertebrates have three RLRs: RIG-1 (encoded by DDX58), MDA5 (IFIH1), and LGP2 (DHX58). Structurally, these are all DEAD-Helicase domain-containing family proteins with a viral RNA binding C-terminal RD (RNA recognition domain), and an N-terminal CARD domain pair that mediates interaction with MAVS (*Loo and Gale, 2011*; *Mukherjee et al., 2014*). Previous phylogenetic studies either did not include RLRs from cartilaginous fishes or have failed to definitively identify each of the three canonical vertebrate RLRs in this lineage, meaning that the ancestral jawed vertebrate RLR repertoire remained unknown (*Krishnaswamy Gopalan et al., 2014*; *Mukherjee et al., 2014*). Our phylogenetic analyses of DEAD-Helicase and CARD domains indicate that orthologs of each of these genes exist in whale shark, revealing that all three RLRs had already diverged in the last common ancestor of extant jawed vertebrates (UFBOOT values all 100; *Table 1*, *Figure 2* and *Figure 2—figure supplement 3*, *Supplementary file 12*, Appendix 3). Further, and consistent with past findings (*Mukherjee et al., 2014*), we found that MDA5 and LGP2 are the result of a vertebrate-specific duplication, while RIG-1 split from these genes much earlier in animal evolution (*Figure 2* and *Figure 2—figure supplement 3*, Appendix 3). We also identified MAVS orthologs in whale shark, elephant shark, and despite difficulties identifying a sequence previously (*Boudinot et al., 2014*), coelacanth (UFBOOT=100; *Figure 2* and *Figure 2—figure supplement 3*, *Supplementary file 12*, Appendix 3). These results show that the mammalian RLR repertoire (and MAVS) was established prior to the emergence of extant jawed vertebrates and has been highly conserved since, consistent with a lack of evidence for large RLR expansions in invertebrates.

TLRs recognize a wide variety of PAMPs and are probably the best known of all innate immune receptors. While large expansions have been observed in several invertebrate lineages (*Huang et al., 2008*; *Rast et al., 2006*), many studies suggest that the vertebrate TLR repertoire is largely conserved (*Boudinot et al., 2014*; *Braasch et al., 2016*; *Wang et al., 2016*). Some teleosts appear to be an exception to this rule; however, this is likely due to the teleost-specific whole-genome duplication, and loss of CD4 and MHC class II in codfishes (*Solbakken et al., 2017*). We identified 13 putative TLRs in whale shark (*Supplementary file 12*; Appendix 3), 11 of which are orthologous to TLR1/6/10, TLR2/28 (x2), TLR3, TLR7, TLR8, TLR9 (x2), TLR21, TLR22/23, and TLR27 (UFBOOT values all ≥99; *Figure 2* and *Figure 2—figure supplement 4*; Appendix 3). The remaining two, along with a coelacanth sequence, represent a novel ancestral jawed vertebrate TLR gene family related to TLR21, for which we propose the name TLR29 (UFBOOT = 99; *Figure 2* and *Figure 2—figure supplement 4*). Thus, the whale shark TLR repertoire is a unique combination when compared

**Table 1.** Vertebrate and invertebrate pathogen recognition receptor (PRR) repertoires.
Superscripts indicate these citations: 1: *Chen et al., 2021*; 2: *Howe et al., 2016*; 3: *Mukherjee et al., 2014*; 4: *Kasamatsu et al., 2010*; 5: *Buckley and Rast, 2015* ; 6: *Tassia et al., 2017*.

| Species | Toll-like receptors (TLRs) | NOD-like receptors (NLRs) | RIG-like receptors (RLRs) |
|---|---|---|---|
| **Jawed vertebrates** | | | |
| *Homo sapiens* (human) | 10 | 21 | 3 |
| *Danio rerio* (zebrafish) | 20[1] | 421[2] | 3[3] |
| *Rhincodon typus* (whale shark) | **13** | **43** | **3** |
| **Jawless vertebrates** | | | |
| *Petromyzon marinus* (lamprey) | 16[4] / 19[5] | 34[5] | 2[3] |
| **Invertebrate deuterostomes** | | | |
| *Ciona intestinalis* | 3[5] | 16[5] | 2[3] |
| *Branchiostoma floridae* | 19[6] / 72[5] | 92[5] | 5[3] |
| *Strongylocentrotus purpuratus* | 104[6] / 253[5] | 203[5] | 6[3] |
| *Cephalodiscus hodgsoni* | 6[6] | | |
| *Ptychodera flava* | 14[6] | | |
| *Saccoglossus kowalevskii* | 10[6] | | 3[3] |
| **Protostomes** | | | |
| *Drosophila melanogaster* | 9[5] | 0[5] | 0 |
| *Daphnia pulex* | 7[5] | 2[5] | |
| *Caenorhabditis elegans* | 1[5] | 0[5] | 2[3] |
| *Capitella teleta* | 105[5] | 55[5] | 2[3] |
| *Helobdella robusta* | 16[5] | 0[5] | 2[3] |
| *Lottia gigantea* | 60[5] | 1[5] | 3[3] |
| **Non-bilaterian animals** | | | |
| *Nematostella vectensis* | 1[5] | 42[5] | 2[3] |
| *Amphimedon queenslandica* | 0[5] | 135[5] | 2[3] |

to all other vertebrates previously studied, being formed from a mix of classic mammalian and teleost TLRs, supplemented with TLR27 and the new TLR29. Our rooted phylogenies indicate that the ancestor of extant vertebrates possessed at least 15 TLRs, while the ancestor of jawed vertebrates possessed at least 19 TLRs (including three distinct TLR9 lineages), both of which are larger repertoires than possessed by modern 2R species (*Figure 2* and *Figure 2—figure supplement 4*; Appendix 3). Unlike invertebrates, where both loss and expansion of TLRs have been extensive, our data suggest that many jawed vertebrate TLRs existed in the jawed vertebrate ancestor, with lineage-specific diversification of jawed vertebrate TLRs primarily resulting from differential loss (as well as genome duplications), supplemented by occasional gene duplication events.

Overall, our findings imply that the jawed vertebrate ancestor possessed a core set of PRRs that has largely shaped the PRR repertoires of modern jawed vertebrates. We propose the budding adaptive immune system formed alongside this core set of PRRs, with concomitant genome duplication-driven expansion of immunoregulatory genes leading these PRRs to become embedded within new, combined innate-adaptive immunity networks. Our results suggest that the impact of this on the propensity for large expansions is PRR type-specific, with expansion of NLRs being recurrent during jawed vertebrate evolution, and massive expansion of TLRs constrained without degeneration of the adaptive immune system. Although reliance upon innate immune receptors is offset in vertebrates due to the presence of the adaptive immune system, our results suggest that differences in PRR repertoires between vertebrates and invertebrates are driven by specific functional needs on a case-by-case basis. Thus, rather than a simple replacement scenario, the interaction with the adaptive immune system, and associated regulatory complexity, is likely a major factor restraining the

proliferation of certain vertebrate PRRs. Although there is clear evidence that PRR expansions can and do occur in jawed vertebrates despite the presence of an adaptive immune system, further in-depth analyses are needed to help better tease out the changes in tempo of PRR diversification across animal phylogeny and whether this associates in any way with the emergence of adaptive immunity.

## Rates of functional genomic evolution and gigantism

Rates of genomic evolution vary considerably across vertebrates, either across clades or in relationship to other biological factors, including body size. We compared rates in two different aspects of genomic evolution with potential functional relationship to gigantism in the whale shark, to those of other vertebrates: rates of amino acid substitution in protein-coding genes and rates of evolution in gene family size.

Substitution rates across a set of single-copy orthologs varied across vertebrate genomes, and these rates were relatively low in the whale shark compared to most other vertebrates (*Figure 3*). We tested for different rates of substitution among vertebrate clades using the two-cluster test implemented by LINTRE (*Takezaki et al., 1995*). Previous use of this test to compare the elephant shark (*C. milii*) genome to other vertebrates determined that *C. milii* has a slower substitution rate than the coelacanth, teleosts, and tetrapods (*Venkatesh et al., 2014*), and that cartilaginous fishes are slower than bony vertebrates (*Hara et al., 2018*). We also found that cartilaginous fishes have a significantly slower rate of substitution (p = 0.0004). In addition, *C. milii* (p = 0.0004) was found to be significantly slower than sharks, consistent with prior work (*Weber et al., 2020*). The whale shark was not significantly different in rate compared to the brownbanded bamboo shark (p = 0.1802), its closest relative included in the analysis. We also found that cartilaginous fishes had a lower rate of molecular evolution compared to subsets of bony vertebrates including ray-finned fishes (p = 0.0004), sarcopterygians (p = 0.001), the coelacanth (p = 0.0004), and tetrapods (p = 0.0088), but not significantly slower than the spotted gar (p = 0.1416). In addition, we found some patterns among other vertebrates consistent with previous studies on rates of molecular substitution, including that ray-finned fishes evolved more rapidly than sarcopterygians (p = 0.0004) and that spotted gar evolved more slowly than teleosts (p = 0.0004).

We then tested whether rates of molecular substitution differed on those branches leading to gigantism among vertebrates. The origins of gigantism in elephants, whales, and the whale shark have previously been shown to correspond to shifts in the rate or mode of body size evolution (*Pimiento et al., 2019*; *Puttick and Thomas, 2015*; *Slater et al., 2017*). We estimated time-varying rates of body size evolution in cartilaginous fishes using BAMM (*Rabosky et al., 2013*). Consistent with previous research (*Pimiento et al., 2019*), we found that gigantism in whale shark corresponds to a discrete shift in the rate of body size evolution to five times the background rate in cartilaginous fishes (Appendix 4; *Appendix 4—figure 1*; *Pimiento et al., 2019*). Using PAML to fit models where the rates of amino acid substitution leading to vertebrate giants to other vertebrates differed, we found this model was significantly different than the strict clock (log-likelihood ratio test p = 1.76 × $10^{-56}$), indicating a significantly different rate of molecular substitution in vertebrate giants. This finding is consistent with earlier evidence that larger-bodied taxa have lower rates of protein evolution (*Martin and Palumbi, 1993*). However, given that the whale shark genome did not appear to evolve significantly more slowly than the brownbanded bamboo shark genome (noted above), or other small-bodied sharks as found previously when focusing on fourfold degenerate sites (*Hara et al., 2018*), there may not be an additive effect on substitution rates in the whale shark genome as both a vertebrate giant and a cartilaginous fish. This implied that substitution rates and body size may have less effect in cartilaginous fishes, which are already overall slowly evolving, in contrast to the pattern seen in other vertebrates.

Rates of change in gene family sizes, due to gain and loss of gene copies within gene families, can also vary across species (*Han et al., 2013*). This represents another potential axis of genomic evolution that may be independent from substitution rates. We estimated rates for gene family size evolution for 10,258 gene families present in the MRCA of vertebrates using CAFE 4.2.1 (*Han et al., 2013*). Average global rates of gene gain and loss in vertebrates were estimated to be 0.0006092 gains/losses per million years. We found that the rate of gene family size evolution in giant vertebrates was significantly faster than in the remaining branches, roughly double the rate in non-giant lineages (p < 0.002). Mean change in gene family size shows that rates of gene family size evolution

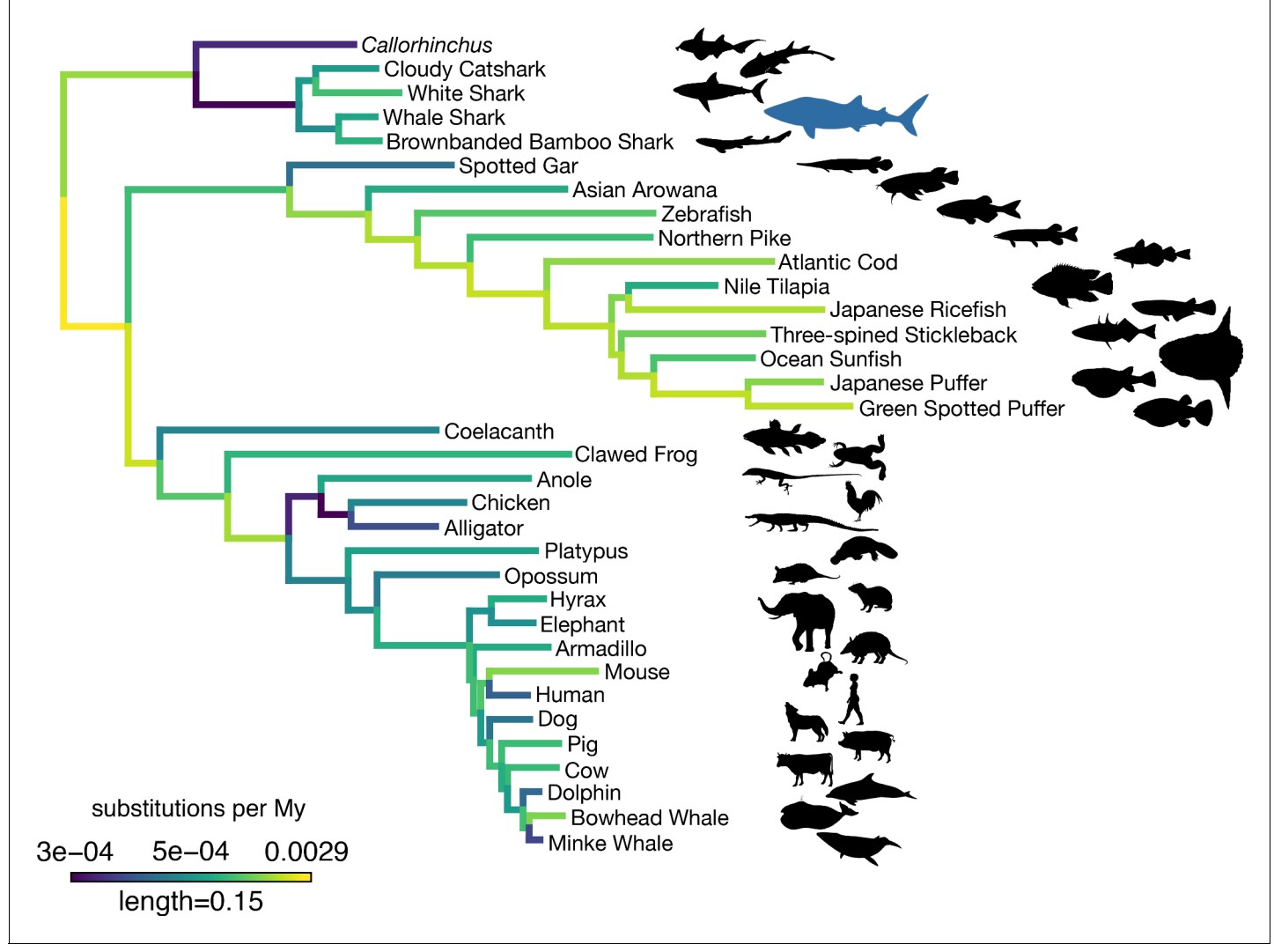

**Figure 3.** Amino acid substitution rate variation among jawed vertebrates. Branches are colored based on rates quantified by substitutions per site per million years of the maximum likelihood tree compared to a time-calibrated tree. Together, sharks have a slower rate of molecular evolution than *Callorhinchus* (see text on two-cluster test). However, sharks do not have a significantly slower rate of molecular evolution than spotted gar. Furthermore, vertebrate giants – including the whale shark, ocean sunfish, elephant, and whales – have significantly lower rates of molecular evolution than other vertebrates. Note, color scale is on normalized reciprocal-transformed data, which emphasizes changes between smaller values of substitution per My.

vary across all taxa including giant lineages, and that an increase in gene family size evolution is not a consistent result of gigantism. However, more complex models in which giant lineages were allowed to vary in rate did not converge. These results suggested that the relationship of gigantism on gene substitution rates do not necessarily predict other forms of genomic evolution, including rates of gene family size evolution.

Replicated shifts in gene gain or loss for specific gene families in independent giant lineages might indicate the consistent effect of selection related to gigantism in particular functional genes. We inferred that 1387 gene families had a rate shift in gene family size evolution on at least one branch in the vertebrate phylogeny (at p < 0.05, *Supplementary file 13*). For those gene families that had a rate shift, on average, around seven independent rate shifts occurred among the vertebrate species considered. Gene families with any shift across the vertebrate phylogeny were enriched for ribosomal genes as well as a few gene families that were enriched for dynein heavy chain genes (*Supplementary file 14*). No gene families independently shifted in all giant taxa exclusive of other vertebrates, and only five gene families independently shifted in any giant taxa

exclusive of other branches in the vertebrate phylogeny (*Supplementary file 13*). This indicates no consistent signal of selection for a rate shift in gene family size evolution for any particular gene family in the evolution of vertebrate giants.

Interestingly, the gene families that shifted in rates of gene gain and loss anywhere in vertebrate phylogeny were also enriched for human orthologs in the Cancer Gene Census (Fisher's exact test, odds ratio = 1.43, p = 0.00414) (*Sondka et al., 2018*), suggesting that these cancer-relevant gene families were also more likely to shift in their expansion/contraction rate across vertebrates. However, this analysis did not consider which branches the shifts occurred on. We hence explored whether gene families that shifted in rate of gene family size evolution across any branch leading to gigantism were enriched for cancer genes. Cancer suppression has evolved by different mechanisms across mammalian lineages, including gene family expansions (*Tollis et al., 2017*); for example, the duplication of tumor suppressor protein TP53 has been implicated in reduced cancer rates in proboscideans (elephants and relatives) relative to other mammals (*Sulak et al., 2016*). By contrast, this same gene family is not expanded in baleen whales (*Tollis et al., 2017*), thus it is already known the same gene families do not expand in all mammalian giants, so we should not expect this to be the case when including fishes. We confirmed a rate shift in gene family size evolution in TP53 in the lineage leading to elephant, but this gene family also had a rate shift for TP53 along the branch leading to minke whale (but not bowhead whale), as well as the non-giant bamboo shark. Therefore, we also tested if all gene families that shifted along a branch leading to a vertebrate giant were enriched for cancer-related genes, including gene families that shifted along non-giant branches. Here, we found that these 1043 gene families were enriched for cancer genes relative to all gene families that shifted in any branch in the vertebrate phylogeny (odds ratio = 2.66, p = 0.00199), with over twice as many cancer-related gene families estimated to have shifted in rate among all vertebrate lineages than the null expectation. That these gene families were not enriched for any particular GO function or protein domain implies that cancer suppression can evolve through various mechanisms. For comparison, we also did the same test but focusing on any gene families that shifted along branches leading to the non-giant vertebrates sister to the giant vertebrates (i.e. brownbanded bamboo shark, pufferfishes, hyrax, bottlenose dolphin), and found that there was no significant enrichment for cancer gene families along those branches (odds ratio = 0.88, p = 0.624). Furthermore, we confirmed the significance of the observed effect size by randomly drawing sets of branches across the vertebrate phylogeny to test for enrichment of cancer genes along random sets of branches, and found that the observed odds ratio of 2.66 was more extreme than 98% of random odds ratios (i.e. p = 0.02). This reinforces that the finding of gene family size evolution shifts along giant branches is significantly enriched for cancer genes.

For the 1387 gene families that had a significant rate shift in gene family size evolution, we then studied if the rates were significantly greater in cancer genes vs. non-cancer genes, depending on whether or not branches led to giant taxa or not. By fitting a linear mixed model using lme4, we found that there was a significantly higher rate of gene family size evolution along branches leading to giant taxa vs. other branches (coefficient 0.0203, p = $5.95e^{-6}$), no effect of whether a gene was related to cancer on rates (coefficient $-0.00245$, p = 0.219), but a significant interaction of cancer suppression function and gigantism (coefficient 0.0102, p = $8.51e^{-5}$), such that rates of gene family size evolution in genes related to cancer leading to giant taxa are even higher than expected relative to the effect of being on a giant branch alone (*Figure 4*). The significantly higher rate of gene family size evolution in vertebrate giants is consistent with the genome-wide patterns estimated above. In these gene families where a rate shift occurred, we found the mean rate of gene family size evolution along branches leading to giant taxa was 3.32 times greater than the mean rate along other branches in cancer genes, but branches leading to giant taxa had only an average rate that was 2.60 times greater than other branches in genes not related to cancer. Overall, this is suggestive that dynamics of vertebrate evolution in cancer-related gene family size among the sampled taxa are driven by the evolution of gigantism.

## Conclusions

As a representative of cartilaginous fishes, a lineage for which only few genomes have thus far been sequenced, the whale shark genome provides an important resource for vertebrate comparative genomics. The new genome assembly based on long reads we reported in this paper represents the best gapless genome assembly thus far among cartilaginous fishes. Comparison of the whale shark

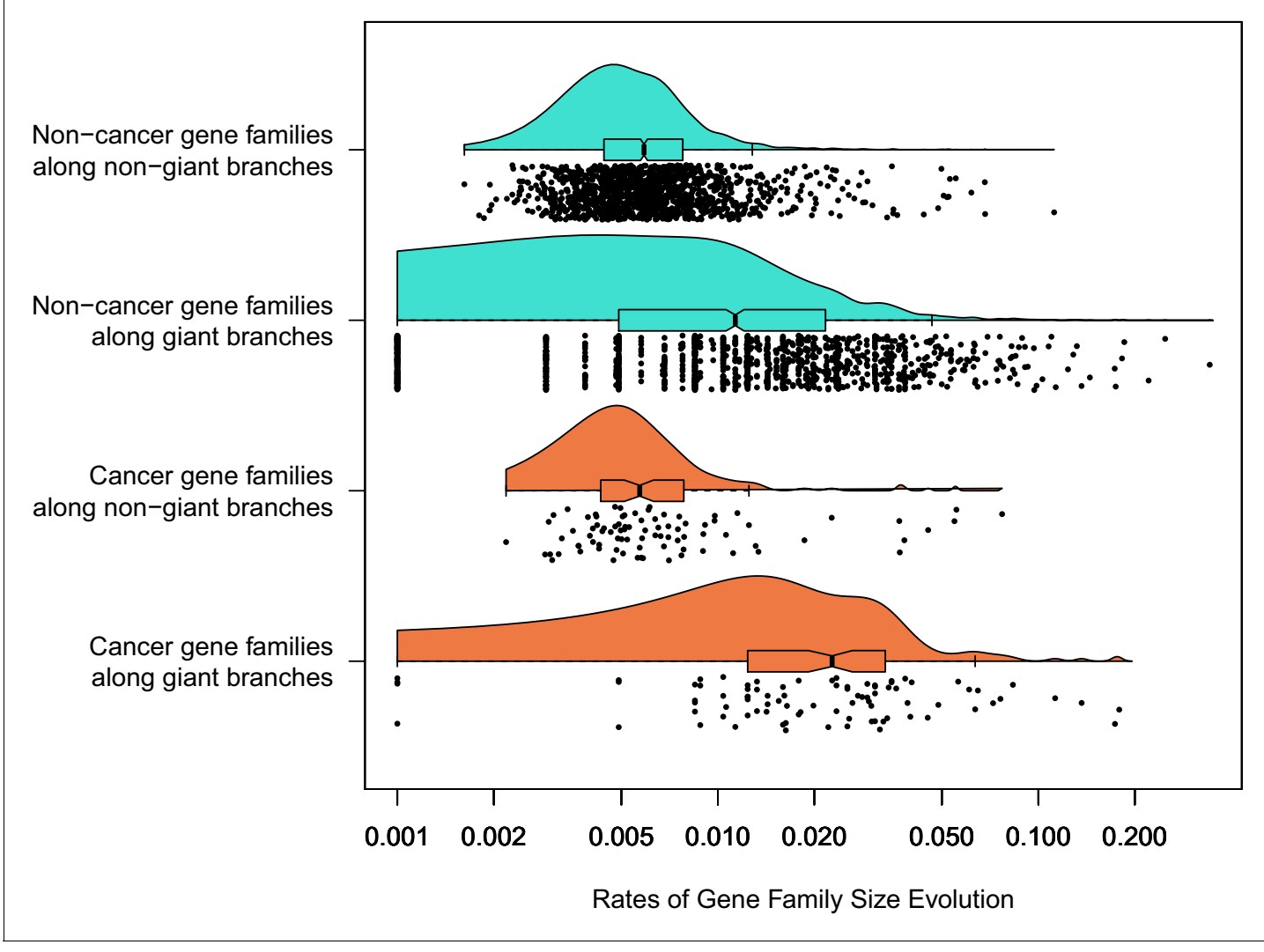

**Figure 4.** Among 1387 gene families with a significant rate shift. Branch-specific rates of gene family size evolution for branches leading to giant taxa were significantly higher than in branches leading to other taxa, and additionally the rate of gene family size evolution was even greater in cancer-related gene families related to other gene families specifically in branches leading to giant taxa.

to other vertebrates not only expands the number of shared gene families that were ancestral to jawed vertebrates but demonstrates that the early vertebrate genome duplications were also accompanied by a burst in the evolution of novel genes. These early gene families are involved in a diversity of functions including reproduction, metabolism, development, and adaptive immunity. We also find differences in gene families implying functional genomic differences between bony vertebrates and cartilaginous fishes, MAS1 and the NK gene cluster. With specific respect to genes involved in innate immune protection, we found divergent patterns of gene gain and loss between NLRs, RLRs, and TLRs, which provide insight into their repertoires in the jawed vertebrate ancestor. These results rejected a scenario where the importance of PRRs is muted in vertebrates by the presence of adaptive immunity, instead indicating the ongoing necessity of ancient PRRs, which were integrated with the new adaptive immune system in the jawed vertebrate ancestor. Finally, we demonstrated that the relationship between rates of gene family size evolution and rates of substitution to gigantism are decoupled, and that gene families that shifted in gene expansion and contraction rate leading to vertebrate giants were enriched for genes with cancer relevance. The whale shark genome helps to build a foundation in shark and vertebrate comparative genomics, which is useful to answer questions of broader vertebrate evolution and convergent evolution of distinctive traits. Further

sequencing of high-quality elasmobranch genomes will continue to enhance research from finding unique, whale shark-specific evolutionary change to illuminating broader patterns of vertebrate evolution.

## Materials and methods

### Genome sequence assembly and assessment

To improve on our earlier efforts to sequence and assemble the whale shark genome (*Read et al., 2017*), we generated PacBio long-read sequences from the same DNA sample. These sequences are available on NCBI SRA under the accession SRX3471980. This resulted in 61.8 Gbp of sequences, equivalent to ~20× fold coverage. The initial assembly was performed using Canu 1.2 (*Koren et al., 2017*) with adjusted parameters to account for the lower input coverage: canu -p asm -d shark genomeSize=3.5g corMhapSensitivity=high corMinCoverage=two errorRate=0.035.

Illumina reads from all paired end read libraries from Read et al. were trimmed using Trimmomatic v0.39 with the following settings: ILLUMINACLIP:adapters.fa:2:30:10 LEADING:5 TRAILING:5 SLIDINGWINDOW:4:15 MINLEN:31 where adapters.fa is a fasta file containing all Illumina sequence adapters packaged with Trimmomatic (*Bolger et al., 2014*). Illumina reads from the single mate pair library from Read et al. were trimmed using NxTrim v0.4.3 using default settings (*O'Connell et al., 2015*). All Illumina reads were aligned to the genome using BWA-MEM (*Li, 2013*) v0.7.12-r1039 with default parameters and alignments were used as input into Pilon v1.18 under default settings (*Walker et al., 2014*) to correct errors in the draft assembly. All reads were then used to scaffold the genome using Platanus 1.2.4 (*Kajitani et al., 2014*). The runs for each library were provided as separate input libraries to platanus scaffold such that the insert sizes will be considered to be different for each library, and the resulting scaffolded assembly was passed to platanus gap_close, both with default settings. Genome size and quality statistics were computed using QUAST v5.0.2 on default settings (*Gurevich et al., 2013*) and compared to the published values from prior studies. The white shark genome paper did not report contig N50; we decomposed the scaffolds into contigs and determined N50 using seqkit v0.10.1 (*Shen et al., 2016*).

We performed *k*-mer analysis using Jellyfish version 2.2.6 on all Illumina reads using the -C setting to count canonical *k*-mers, a *k*-mer size (-m) of 21 and hash size (-s) of 100M (*Marçais and Kingsford, 2011*). Then, we used GenomeScope to fit a model that allows for assembly-free estimation of genome size, heterozygosity, and repeat content (*Vurture et al., 2017*), providing the *k*-mer size of 21, read length of 100, and using the default maximum *k*-mer coverage of 1000 (GenomeScope accessed June 5, 2017). We used KAT v2.2.0 to plot the *k*-mers and visualize the copy number in the genome of *k*-mers in the raw read Illumina data (*Mapleson et al., 2016*). We first used kat comp to compare the Jellyfish *k*-mer counts to the genome assembly, and plotted the results using kat plot spectra-cn.

We assessed gene completeness with conserved vertebrate orthologs using BUSCO v2 (*Simão et al., 2015*) and CVG orthologs using gVolante (version 1.2.1; accessed April 23, 2019) (*Hara et al., 2015*), and by mapping RNA-seq reads (Appendix 1). The trimmed reads were then reused to call SNPs to assess heterozygosity using freebayes under default settings (*Garrison and Marth, 2012*). We then used vcflib packages vcffilter to filter the results for a minimum quality of 20 (-f 'QUAL >20') and vcfstats to count the number of SNPs (*Garrison et al., 2021*).

### Transcriptome sequencing

Approximately 30 million short-read pairs for whale shark transcripts were obtained with paired-end 127 cycles from blood cells of a male and a female by the Illumina HiSeq 1500, as described previously (*Hara et al., 2018*). Animal handling and sample collections at Okinawa Churaumi Aquarium were conducted by veterinary staff without restraining the individuals, in accordance with the Husbandry Guidelines approved by the Ethics and Welfare Committee of Japanese Association of Zoos and Aquariums. Downstream handling of nucleic acids was conducted in accordance with the Guideline of the Institutional Animal Care and Use Committee (IACUC) of RIKEN Kobe Branch (Approval ID: H16-11). Transcriptome sequence data are available at NCBI BioProject ID PRJDB8472 and DDBJ DRA ID DRA008572.

## Gene prediction

Genes predictions were provided to us by RefSeq using their genome annotation pipeline version 7.3 (*Warren et al., 2017*), details of the resulting annotation are publicly available (*Rhincodon typus Annotation Report, 2018*). This annotation included alignments of RNAseq data from gray bambooshark *Chiloscyllium griseum* kidney and spleen, nurse shark *Ginglymostoma cirratum* spleen and thymus, and brownbanded bambooshark *C. punctatum* retina, as well as protein alignments from Actinopterygii, and RefSeq protein sequences for Asian arowana *Scleropages formosus*, coelacanth, spotted gar, zebrafish, clawed frog, and human. After preliminary orthology determination, we determined additional genes absent in whale shark conserved among vertebrates, which we annotated by aligning protein sequences from these genes from human, gar, coelacanth, and mouse to whale shark using genBLAST v1.39 (*She et al., 2011*) with these settings: -p genblastg -e 1e-5 -g T -gff -cdna -pro -pid (Appendix 1, *Supplementary file 15*), and for elephant shark (*Supplementary file 16*).

## Orthology inference

We identified orthologs from the whale shark genome by comparison to publicly available chordate genomes. We compiled chordate proteomes for 32 species representing major vertebrate clades, the sea squirt *Ciona intestinalis*, and lancelet *Branchiostoma floridae* (*Supplementary file 3*). In selecting representative vertebrates, we specifically included the ocean sunfish, African elephant, and two baleen whale genomes (minke whale, bowhead whale), and the most closely related genomes available for these taxa (*Takifugu rubripes* and *Dichotomyctere nigroviridis*, rock hyrax, and bottlenose dolphin). These ortholog clusters were used for the identification of origins of gene families in chordate evolution and genes that originated in the MRCA of jawed vertebrates, studying enrichment or changes in functional annotation associated with these orthogroups, phylogenomics, estimation of rates of molecular substitution, and estimation of rates of gene duplication and loss.

Ortholog clusters from proteomes were determined using OrthoFinder v2.2.6 with default settings (*Emms and Kelly, 2015*). With the resulting hits, OrthoFinder adjust scores for reciprocal best hits while accounting for gene length bias and phylogenetic distance, then proceeds with clustering genes into orthogroups. Preliminary orthology determination suggested many potential missing orthologs in the elephant shark and whale shark genomes. We thus performed orthology-based annotation using genBLAST (*She et al., 2011*) as noted under Gene prediction section in Materials and methods, added newly identified proteins to the proteomes of whale shark and *Callorhinchus*, and reran the OrthoFinder pipeline including these proteins.

All proteins were then annotated for GO and Pfam terms using InterProScan 5.32–71.0 (*Jones et al., 2014*), and representative annotations were assigned to each chordate orthogroup using KinFin 1.0 with the `--infer-singletons` option on to interpret gene families absent from clusters as singletons, then running the functional_annotation_of_clusters.py script packaged with KinFin under default settings, which assigns an annotation to a gene family if at least 75% of proteins in the gene family has that annotation, and at least 75% of taxa within the cluster have a protein with that annotation (*Laetsch and Blaxter, 2017*; *Supplementary file 5*).

## Gene family origin and loss

A custom R script is provided for analyses run for this section (*Source code 1*). To infer when gene families (as inferred from OrthoFinder) were gained and lost in vertebrate evolution, we mapped the origins and losses of gene families to the species tree parsimoniously, assuming that gene families have a single origin, but can be lost (*Laetsch and Blaxter, 2017*). We were then able to count the number of gene families present at the MRCA of nodes, the number of novel gene families that originated along each branch, and the number of gene families lost along each branch (including gene families uniquely lost along each branch). We also determined the number of gene families conserved in all descendants (core genes) and the number of novel gene families conserved in all descendants (novel core genes).

We aimed to confirm that the number of novel gene families in the jawed vertebrate ancestor was not inflated by artifactual oversplitting of ohnologs (gene duplicates that arose from 2R of whole- genome duplication early in vertebrate evolution). *Singh et al., 2015*, independently used a synteny-aware method to identify ohnologs in a subset of vertebrate genomes. We compared our

assignment of human orthologs to gene families to the assignment of human orthologs to Singh et al. ohnolog families (downloaded June 2, 2020). We determined whether our gene families and Singh et al. ohnolog families matched and whether human orthologs were assigned to a single gene family or ohnolog. We replicated this for green anole, spotted gar, zebrafish, and possum ohnologs. To find the common genes between the Ensembl protein IDs we clustered and Ensembl gene IDs provided by Singh et al., we used the R package biomaRt v2.45.2 to translate identifiers (*Durinck et al., 2005*). Scripts for this analysis are provided (*Source code 2*).

Based on the representative annotations for each orthogroup determined above, we then determined whether groups of gene families that were gained or lost along branches in the vertebrate phylogeny were enriched for certain functions using a Fisher's exact test. Within each comparison, we adjusted the p-value to correct for multiple hypothesis testing by the Benjamin-Hochberg (BH) method using the p.adjust function in R (*Wright, 1992*). Corrected p-values under the BH method can be interpreted at a significance threshold that is equivalent to the false discovery rate. We considered functions enriched with an adjusted p-value of 0.05 and false discovery rate of 0.05.

## Innate immunity analyses

### Homology identification

Sequence similarity searches were performed using BLAST+ v2.6.0 to identify putative homologs of TLRs, NLRs, and RLRs (*Altschul et al., 1990*). An alternative approach using profile hidden Markov models, HMMER (version 3.1) (*Eddy, 1998*), was also tested for TLRs; the results obtained were identical, except that BLAST returned an additional putative TLR. Due to this, HMMER results were not applied in subsequent analyses, and HMMER was not applied elsewhere (*Eddy, 1998*). Searches for whale shark TLR and RLR homologs were performed using all other sequences present in the TLR and RLR trees. Retention of sequences for further analyses was reliant on a reciprocal blast hit to a TLR or an RLR in the Swissprot reviewed database or the NCBI non-redundant protein set (*Boeckmann et al., 2003*).

For NLRs, detection is more complicated, as some NLRs do not contain computationally detectable NACHT domains (i.e. some family members, even in humans, are false negatives in domain-based search tools and databases), despite the NACHT domain being the defining feature of NLR family members. Further, some of these genes contain other domains and are also included in other gene families where most members do not contain NACHT domains. As such, for the main analysis performed here, those sequences in the predicted proteome and translated transcriptome containing a predicted NACHT domain according to the NCBI CD-search webserver (*Marchler-Bauer et al., 2015*) are noted as such (and should be considered as the conservative set of whale shark NLR-like sequences). Additional sequences from the predicted protein set with a blast hit to known NLRs were also included to permit detection of potential orthologs of NLRs not found in the conservative set with definite/detectable NACHT domains. Proteins containing the closely related NB-ARC domain were also extracted from the whale shark proteome.

In cases where a transcript matches the genomic location of a predicted protein, the predicted protein is the sequence reported. Where multiple predicted proteins refer to the same genomic location, only a single sequence is retained for further analysis.

### Phylogenetic datasets

For NLRs, we performed phylogenetic analyses of the whale shark putative NLRs to known NLRs from human and zebrafish, both of which are highly phylogenetically relevant and well studied in this regard. Proteins containing the closely related NB-ARC domain were used as an outgroup in NLR analyses, along with human APAF-1 which also harbors an NB-ARC domain (*Urbach and Ausubel, 2017*).

To better understand RLR and MAVS evolution, we used two datasets. The first of these was based on the central DEAD-Helicase domains (hence, excluding MAVS) to define which of the three RLR proteins could be found in whale shark, and infer the jawed vertebrate RLR repertoire, also following *Mukherjee et al., 2014*. For the RLR datasets, members of each of the three vertebrate RLR families, some invertebrate RLRs, and a selection of DICER proteins sequences as an outgroup (*Mukherjee et al., 2014*) were gathered to generate a phylogenetically informative dataset (i.e. aiming to include representatives of each of the major vertebrate classes for which genome data were

available). Full-length proteins were aligned for phylogenetic analysis of DEAD-Helicase domains, and trimmed to the start and end of these domains based on the three human RLR sequences (*Mukherjee et al., 2014*). The second dataset was based on individual CARD domains, as the presence of two CARD domains in RIG-1 and MDA5 is thought to have come about through independent domain duplication in each lineage, which would mislead phylogenetic analyses if ignored (*Korithoski et al., 2015*). The same process as for DEAD-Helicase domains was performed for the CARD domains (*Korithoski et al., 2015*).

For the TLR dataset, a large set of TLR nucleotide sequences were taken from a past study that densely sampled vertebrates (*Wang et al., 2016*; *Supplementary file 17*). TLR sequence from gray bamboo shark (*C. griseum*) was also included (*Krishnaswamy Gopalan et al., 2014*). Following trimming, the alignment consisted almost entirely of sites from the TIR domain, so TIR domains were not specifically extracted for this analysis.

For the NLR analysis, the described set of human NLRs and NACHT domain containing proteins, as well as the closely related NB-ARCs as an outgroup (*Urbach and Ausubel, 2017*), were downloaded from NCBI protein database. Sequences of zebrafish, where NLRs are massively expanded (*Howe et al., 2016*; *Stein et al., 2007*), were also included in this analysis, but these were downloaded from the InterPro website (i.e. all *Danio rerio* proteins containing a NACHT domain) (*Hunter et al., 2009*). A very large number of zebrafish sequences were obtained, so to reduce the prevalence of pseudo-replicate sequences (that are likely to be uninformative in the context of understanding the whale shark NLR repertoire), CD-HIT (*Fu et al., 2012*) was used to cluster zebrafish sequences with greater than 75% identity prior to phylogenetic analysis. An additional NLR analysis was performed focusing specifically on NOD1s, this employed NOD2 as an outgroup based on the larger NLR analysis and included NOD1s identified by BLAST in elephant shark. Notably, our NLR analysis relies on poorer taxon sampling compared to that for the RLR and NLR datasets. This is due to a relative paucity of previously characterized NLR repertoires across vertebrate species. Importantly, although this does not lend itself well to understanding the tempo of lineage-specific gene family expansion/contraction, it does not preclude detection of such events along the lineages leading to the species included in the analysis.

## Multiple sequence alignment and phylogenetic analyses

Multiple sequence alignments were generated with MAFFT (version 7.313) (*Katoh and Standley, 2013*) using default parameters for the larger TLR and NLR datasets, but using the more intensive L-INS-i method for RLRs and the focused NOD1 NLR dataset. trimAl (version 1.2rev59) (*Capella-Gutierrez et al., 2009*) was applied to remove gap-rich sites, which are often poorly aligned, from the alignments using the 'gappyout' algorithm. BMGE (version 1.12) (*Criscuolo and Gribaldo, 2010*) was then used to help minimize the number of saturated sites in the remaining alignment (as identified using the BLOSUM30 matrix). The RLR analyses were not subjected to this BMGE analysis, as these were derived from conserved domains (meaning that alignments were based on relatively conserved sequence tracts and were already quite short). The NOD1-focused NLR alignments were judged to contain relatively similar sequences and were not subjected to either trimAl or BMGE analyses. Phylogenetic analyses were performed in IQ-TREE (version: omp-1.5.4) (*Nguyen et al., 2015*) using 1000 ultrafast bootstrap replicates (*Minh et al., 2013*) and the best-fitting model of amino acid substitution. Best-fitting substitution models were determined according to the Bayesian information criterion with ModelFinder from the IQ-TREE package (*Kalyaanamoorthy et al., 2017*), and ultrafast bootstrap support was computed to assess branch support (*Hoang et al., 2018*). The following (best-fitting) models were applied for each dataset: LG+I+G for RLR CARD domains dataset, LG+I+F+G for RLR DEAD-Helicase domains dataset, JTT+I+F+G for the TLR dataset, JTT+F+G for the NLR dataset, and JTT+I+F+G for the NOD1-focused NLR dataset. The trees were rooted either by outgroup, or the TLR tree was rooted minimal ancestor deviation method (*Tria et al., 2017*). This is unlike many other TLR trees produced in previous studies which are unrooted (*Roach et al., 2005*; *Wang et al., 2016*).

## Phylogenomics

Orthogroups were filtered to single-copy orthologs for phylogenomic analyses. We determined orthologs from orthogroups by reconstructing orthogroup trees and used tree-based orthology

determination using the UPhO pipeline (*Ballesteros and Hormiga, 2016*). The paMATRAX+ pipeline bundled with UPhO was used to perform alignment (mafft version 7.130b), mask gaps (trimAl version 1.2), remove sequences containing too few unambiguous sites, and check that the minimum number of taxa are present (using the Al2Phylo script part of UPhO), and then reconstruct phylogenies (IQ-TREE v1.6.10) (*Capella-Gutierrez et al., 2009*; *Katoh and Standley, 2013*; *Nguyen et al., 2015*). Next, we used UPhO to extract orthologs by identifying all maximum inclusive subtrees from orthogroups with at least five species, with the allowance for in-paralogs (paralogs that arose after all species divergences in the phylogeny, and thus do not affect relative relationships in the phylogeny), and retained the longest in-paralogous sequence for each species within each ortholog. For each single-copy ortholog, we aligned, trimmed, and sanitized sequences using the paMATRAX+ pipeline.

To select the most reliable sequences for inferring a phylogenomic time tree, we filtered for the most informative loci using MARE version 0.1.2-rc with default settings except -t (taxon weight) set to 10 to weight the retention of taxa higher than retaining loci in the alignment (*Misof et al., 2013*). Next, orthologs without lamprey, *Callorhinchus*, whale shark, *Branchiostoma*, and *Ciona* were excluded. We also filtered down to loci that supported the monophyly of vertebrate, gnathostome, chondrichthyan, and osteichthyan clades. After our filtering we were left with an alignment comprising 281 loci and 209,275 residues. We concatenated the sequences and selected the best model of amino acid substitution and partitioning scheme and inferred a maximum likelihood phylogeny using IQTREE v1.6.10 (*Hoang et al., 2018*; *Kalyaanamoorthy et al., 2017*; *Nguyen et al., 2015*) with the followings settings: -bb 1000 -bnni -m MFP+MERGE -rcluster 10. The tree was rooted using the amphioxus *Branchiostoma*. The phylogeny was largely consistent with consensus arising from phylogenomic studies. We also inferred a phylogeny accounting for incomplete lineage sorting using ASTRAL v5.7.1 (*Zhang et al., 2018*) based on gene trees (not shown), which was identical in topology except for the placement of armadillo (Xenarathra) sister to Afrotheria in the ASTRAL tree vs. armadillos sister to Boreoeutheria in IQ-TREE. This relationship has historically been difficult to reconstruct and is consistent with prior conflicts between concatenated and coalescent-based analysis on the placement of Xenarthra with far more taxa (*Esselstyn et al., 2017*). In addition, none of our focal results are reported within mammals, where the relationships of Xenarthrans could be relevant.

Numerous fossil-based node calibrations were identified from the literature. Most node ages were derived from age ranges published in the Fossil Calibration Database (*Benton et al., 2014*) and are listed in *Supplementary file 18*. While previously the age of crown Chondrichthyes (here, the MRCA of Holocephali + Elasmobranchii) has been suggested to range from 333.56 to 422.4 Ma, the minimum age was recently pushed further back to 358 Ma based on multiple holocephalan fossils (*Coates et al., 2017*). To assess the concordance of the fossil calibrations, we used treePL version 1.0 to estimate divergence times from the ML tree with each fossil calibration using penalized likelihood, then performed cross-validation and evaluated the concordance of the fossils to the time tree to identify and exclude outliers (*Near et al., 2005*). After excluding fossils that were discordant with the others, we estimated divergence times using treePL with the remaining fossil calibrations. The final treePL config file is provided (*Source code 3*).

## Tests for rates of substitution

Based on our aligned matrix from single-copy orthologs used for phylogenomics, we tested for differences in rates of molecular substitution between vertebrates by using the two-cluster test implemented in LINTRE (April 17, 2010 version) (*Takezaki et al., 1995*), using amino acid p-distances between taxa to estimate branch lengths. The two-cluster test is designed to test if the rates in two clades are significantly different by comparison to an outgroup. We tested rates on the full tree, as well as focused on certain cluster pairs by subsetting the dataset to focus on specific clades for comparison. Sequences were converted to phylip format from fasta format using pxs2phy using phyx v1.01 (*Brown et al., 2017*). Scripts to implement tests run are provided (*Source code 3*).

We also compared rates of genomic evolution of four independent instances of vertebrate gigantism (whale shark, elephant, baleen whales, ocean sunfish) relative to the background rate of molecular evolution among vertebrates. To do this, we used PAML 4.9i to compute the likelihoods of the alignment of single-copy orthologs used for phylogenomics under two different models of molecular evolution (*Yang, 2007*). We computed the likelihood of the data under a strict clock model (single-

rate model) and under a local clock model (two-rate model) where the clock rate differed on branches leading to vertebrate giants. We then determined significance using the likelihood ratio test. PAML control files are provided (*Source code 3*).

## Rates of gene family size evolution

We estimated rates of gene family expansion and contraction across vertebrates among gene families. OrthoFinder output includes counts of the size of each orthogroup (i.e. gene families) for each species. We analyzed the evolution of gene family size under a birth-death process using CAFE version 4.2.1 (*Han et al., 2013*), with the gene family size evolutionary rate parameter λ. We focused on gene families present in the MRCA of vertebrates and filtered these only to gene families present in at least two species, and to exclude gene families that exceed 100 copies in any species, as large gene families have too large variance for consistent rate parameter estimation, resulting in 10,258 gene families. We used the caferror.py script to estimate species-by-species error rates in the annotation to improve the accuracy of rate estimation (*Han et al., 2013*). We used a time-calibrated phylogeny of vertebrates for this analysis (see above). We provide scripts used for running CAFE (*Source code 4*).

We estimated rates of gene duplication and loss across vertebrates under a single λ model, and two multi-λ models: a two λ model where branches leading to gigantism had a second rate, and a five λ model where the rate categories were the background and a separate rate for each of the four independent origins of gigantism. However, the five λ model did not converge. To test for significance of the observed difference in likelihoods between the two λ model and the single λ model, we simulated gene family evolution with 500 replicates under these models and estimated the log-likelihood ratios from this null, simulated distribution. The p-value corresponds to the proportion of simulated replicates which had a smaller log-likelihood ratio than observed. When fitting the λ model, CAFE 4 additionally computes rates of duplication and loss along each branch for each gene family and tests whether significant rate shifts occur along each branch (*Source code 4*). p-Values < 0.05 indicate a significant rate shift in gene family size evolution rate.

We identified gene families that had shifted and tested whether they were enriched for GO and Pfam terms (as above). We also tested for enrichment of gene families including human orthologs related to cancer. Cancer-related gene families were determined by downloading the gene families from the COSMIC Cancer Gene Census (*Sondka et al., 2018*) and determining which orthogroups included the human ortholog based on the Ensembl gene identifier provided by the CGC (database version 91, April 07, 2020, accessed June 3, 2020). Ensembl gene ENSG identifiers were matched to the Ensembl protein ENSP identifiers (which we used for orthogroup determination) using biomaRt version 2.45.2, database accessed September 9, 2020 (*Durinck et al., 2005*).

We also tested whether gene families that shifted in expansion and contraction rate along branches leading to giant taxa were enriched for cancer genes. Six branches were tested in this set of focal branches relative to all other branches in the vertebrate phylogeny: the branch leading to whale shark, the branch leading to ocean sunfish, the branch leading to African elephant, and the branches corresponding to the clade of baleen whales (the clade sister to bottlenose dolphin). To confirm whether the results were more extreme than expected, we performed two tests. First, we drew the branches corresponding to the non-giant sister taxa of the vertebrate giants, then tested for whether these were enriched for cancer genes. Second, we tested for cancer gene enrichment on 100 permutations of selecting six random branches without replacement from across the vertebrate phylogeny. We then compared the observed odds ratio of enrichment for cancer genes to this null distribution.

We then compared the rate of gene family size evolution for gene families related to cancer to rates of gene families not related to cancer along branches leading to giant vertebrates and the remaining branches in phylogeny. Branch-wise rates of gene family size evolution were estimated by computing the difference in estimated ancestral and descendant gene family sizes of each branch and dividing by time. We also used the lme4 package and lmerTest packages to fit a linear mixed model and test for significant contribution on rate depends on whether it was estimated for a cancer gene or not, whether the rate was estimated on a branch leading to a giant taxon or not, the interaction of these variables, and with gene family as a random effect (*Bates et al., 2015*; *Kuznetsova et al., 2017*) .

## Acknowledgements

The sequencing service was provided by the Norwegian Sequencing Centre (https://www.sequencing.uio.no/), a national technology platform hosted by the University of Oslo and supported by the 'Functional Genomics' and 'Infrastructure' programs of the 'Research Council of Norway and the Southeastern Regional Health Authorities'. We thank F Thibaud-Nissen for assistance with genome annotation through NCBI RefSeq. We thank B Morgan and the High Performance Computing oversight committee for access and assistance with the Center for Advanced Science Innovation and Commerce (CASIC) supercomputer at Auburn University, RA Petit III for assistance with computing at Emory University, and High Performance Computing at George Washington University for assistance with Colonial One and Pegasus. We thank the staff of Laboratory of Phyloinformatics in RIKEN BDR for transcriptome sequencing. Inference of gigantism in the whale shark was made possible by body size data kindly provided by C Mull. We are thankful for funding provided from the Georgia Aquarium and the Emory School of Medicine Development. S Koren and AM Phillippy were supported by the Intramural Research Program of the National Human Genome Research Institute, National Institutes of Health. Silhouettes used throughout via PhyloPic: *Callorhinchus*, CC BY-SA by Milton Tan, originally by Tony Ayling; whale shark, CC BY-SA by Scarlet23 (vectorized by T Michael Keesey); spotted gar, tilapia, stickleback, CC BY-NC-SA by Milton Tan; Asian arowana, coelacanth, CC BY-NC-SA by Maija Karala; clawed frog, anole, platypus, opossum, elephant, CC BY by Sarah Werning; alligator, CC BY-NC-SA by Scott Hartman; mouse, CC BY-SA by David Liao; dolphin, CC BY-SA by Chris Huh; catshark, brownbanded bamboo shark, white shark, zebrafish, pike, cod, ricefish, ocean sunfish, chicken, armadillo, hyrax, human, dog, pig, cow, minke whale, bowhead whale, public domain.

## Additional information

### Competing interests

Shigehiro Kuraku: Reviewing editor, *eLife*. The other authors declare that no competing interests exist.

### Funding

| Funder | Author |
| --- | --- |
| Georgia Aquarium | Alistair DM Dove |
| School of Medicine, Emory University | Timothy Read |

The funders had no role in study design, data collection and interpretation, or the decision to submit the work for publication.

### Author contributions

Milton Tan, Conceptualization, Data curation, Formal analysis, Visualization, Methodology, Writing - original draft, Project administration, Writing - review and editing, Implemented analyses for genome polishing, phylogenomics, orthology determination, gene family evolution, RNAseq mapping; Anthony K Redmond, Conceptualization, Formal analysis, Visualization, Methodology, Writing - original draft, Writing - review and editing, Performed analyses and wrote the results and methods sections on innate immune genes; Helen Dooley, Conceptualization, Formal analysis, Methodology, Writing - original draft, Writing - review and editing, Performed analyses and wrote the results and methods sections on innate immune genes; Ryo Nozu, Resources, Investigation, Writing - review and editing, Generated RNAseq data; Keiichi Sato, Resources, Investigation, Writing - review and editing; Shigehiro Kuraku, Resources, Data curation, Investigation, Writing - review and editing; Sergey Koren, Data curation, Formal analysis, Writing - review and editing, Assembled genome using Canu; Adam M Phillippy, Resources, Data curation, Formal analysis, Writing - review and editing, Assembled genome using Canu; Alistair DM Dove, Resources, Supervision, Funding acquisition,

Investigation, Writing - review and editing; Timothy Read, Conceptualization, Resources, Supervision, Funding acquisition, Investigation, Writing - review and editing

## Author ORCIDs
Milton Tan  https://orcid.org/0000-0002-9803-0827
Helen Dooley  http://orcid.org/0000-0002-2570-574X
Ryo Nozu  http://orcid.org/0000-0002-1099-3152
Shigehiro Kuraku  http://orcid.org/0000-0003-1464-8388
Alistair DM Dove  http://orcid.org/0000-0003-3239-4772

## Decision letter and Author response
Decision letter https://doi.org/10.7554/eLife.65394.sa1
Author response https://doi.org/10.7554/eLife.65394.sa2

# Additional files

## Supplementary files
• Source code 1. Scripts for assessing gene family gain and loss and enrichment of gene family functional annotations (R).

• Source code 2. Scripts for comparing gene family assignment to known ohnologs (ZIP).

• Source code 3. Scripts for estimating divergence times using TREEPL and comparing rates of substitution using LINTRE and PAML (ZIP)) *Bates et al., 2015*, *Kuznetsova et al., 2017*.

• Source code 4. Scripts for summarizing CAFE results for rates of gene family size evolution and enrichment of functional annotations and cancer-related function (ZIP).

• Source code 5. Scripts for annotating repetitive sequences (SH).

• Source code 6. Scripts for assessing rates of body size evolution across cartilaginous fishes and compared to the whale shark (R).

• Source data 1. Table for Statistical Reporting Form.

• Supplementary file 1. Comparison of whale shark genome assemblies.

• Supplementary file 2. BUSCO v2 and core vertebrate gene (CVG) results. BUSCO v2 and CVG results for brownbanded bamboo shark and cloudy catshark were those reported by *Hara et al., 2018 Figure 2—figure supplement 1d*, who did not report Complete Single-copy and Complete Duplicate numbers and only reported percentages. *Callorhinchus* CVG scores are reported on the gVolante database (https://gvolante.riken.jp/script/database.cgi, accessed January 19, 2021). The BUSCO v2 set has 2586 vertebrate orthologs, while the CVG has 233 total genes. Note CVG does not report if complete are single copy or duplicated. Percentages in parentheses. Note also that *Callorhinchus* was used in the ortholog design of both sets and therefore BUSCO and CVG overestimate its completeness.

• Supplementary file 3. Chordate species with whole-genomic data included in comparative genomic analyses.

• Supplementary file 4. Orthogroup assignment by OrthoFinder of chordate proteins (CSV).

• Supplementary file 5. Gene ontology (GO) and Pfam annotations of orthogroups assigned by KinFin (tab-delimited table TSV).

• Supplementary file 6. Summary of functional enrichment tests of gene families gained and lost throughout chordate evolution. We tested whether or not gene families in the foregrounds were enriched for functional terms and domains (gene ontology [GO], Pfam) relative to the background of what was present at a relevant ancestor.

• Supplementary file 7. Significantly enriched functional and domain terms identified in novel gene families (orthogroups, *Supplementary file 5*) gained in the most recent common ancestor (MRCA) of Olfactores. n refers to the number of these gene families with that function gained. p refers to uncorrected p-values for Fisher's exact test, adj.p refers to the adjusted p-value for multiple testing

(see Materials and methods). See *Supplementary file 19* for specific assignments of human gene names to each orthogroup.

• Supplementary file 8. Significantly enriched functional and domain terms identified in novel gene families (orthogroups) gained in the most recent common ancestor (MRCA) of vertebrates. n refers to the number of these gene families with that function gained. p.value refers to uncorrected p-values for Fisher's exact test, adj.p refers to the adjusted p-value for multiple testing (see Materials and methods). See *Supplementary file 19* for specific assignments of human gene names to each orthogroup.

• Supplementary file 9. Significantly enriched functional and domain terms identified in novel gene families (orthogroups) gained in the most recent common ancestor (MRCA) of gnathostomes. n refers to the number of these gene families with that function gained. p.value refers to uncorrected p-values for Fisher's exact test, adj.p refers to the adjusted p-value for multiple testing (see Materials and methods). See *Supplementary file 19* for specific assignments of human gene names to each orthogroup.

• Supplementary file 10. Significantly enriched functional and domain terms identified in novel gene families (orthogroups) gained in the most recent common ancestor (MRCA) of Osteichthyes. n refers to the number of these gene families with that function gained. p.value refers to uncorrected p-values for Fisher's exact test, adj.p refers to the adjusted p-value for multiple testing (see Materials and methods). See *Supplementary file 19* for specific assignments of human gene names to each orthogroup.

• Supplementary file 11. Significantly enriched functional and domain terms identified in gene families (orthogroups) lost in the most recent common ancestor (MRCA) of Chondrichthyes. n refers to the number of these gene families with that function gained. p.value refers to uncorrected p-values for Fisher's exact test, adj.p refers to the adjusted p-value for multiple testing (see Materials and methods). See *Supplementary file 19* for specific assignments of human gene names to each orthogroup.

• Supplementary file 12. Whale shark pathogen recognition receptor (PRR) gene accessions. Sequences that have identical or are isoforms of the same gene are indicated. TLR9 and TLR29 sequences that were not annotated are also indicated.

• Supplementary file 13. CAFE output for rates of gene duplication and loss of vertebrate orthogroups computed under a single global rate of gene duplication and loss for orthogroups (TXT).

• Supplementary file 14. Significantly enriched functional and domain terms identified in gene families (orthogroups) with a rate shift in gene family size in any part of the vertebrate phylogeny. n refers to the number of these gene families with that function gained. p.value refers to uncorrected p-values for Fisher's exact test, adj.p refers to the adjusted p-value for multiple testing (see Materials and methods). See *Supplementary file 19* for specific assignments of human gene names to each orthogroup.

• Supplementary file 15. Putative conserved vertebrate genes absent from the whale shark RefSeq annotation that were annotated using genBlast. Annotations are for the GCF_001642345.1 genome assembly (GFF).

• Supplementary file 16. Putative conserved vertebrate genes absent from the *Callorhinchus* RefSeq annotation that were annotated using genBlast. Annotations are for the GCF_000165045.1 genome assembly (GFF).

• Supplementary file 17. Species included and excluded for toll-like receptor (TLR) analysis from *Wang et al., 2016* dataset (XLSX).

• Supplementary file 18. Fossil calibration age ranges, and the result of fossil concordance analysis. Discordant fossils were excluded from divergence time analysis. All age ranges are derived from *Benton et al., 2014*, except for the age of Chondricthyes, which were derived from *Coates et al., 2017*.

- Supplementary file 19. Human gene names of human orthologs assigned to each orthogroup (TXT).

- Supplementary file 20. Repeat library annotated using the MAKER repeat annotation pipeline (FASTA). Repeat classification of each repeat sequence follows a '#' delimiter.

- Supplementary file 21. Repetitive element content of the whale shark genome assembly (for methods, see Appendix 1).

- Supplementary file 22. Whale shark transcriptome annotation based on StringTie (GFF).

- Transparent reporting form

## Data availability

Raw genome sequencing data have been deposited to SRA under SRX3471980. Raw transcriptome sequence sequence data are available at NCBI BioProject ID PRJDB8472 and DDBJ DRA ID DRA008572. The assemblies have been deposited to GenBank; the contig assembly is accessioned as GCA_001642345.2, and the scaffold assembly is accessioned as GCA_001642345.3.

The following datasets were generated:

| Author(s) | Year | Dataset title | Dataset URL | Database and Identifier |
|---|---|---|---|---|
| Tan M, Read TD, Dove ADM | 2019 | Whole genome sequencing of the Whale Shark (Rhincodon typus) | https://www.ncbi.nlm. nih.gov/sra/SRX3471980 | NCBI Sequence Read Archive, SRX3471980 |
| Nozu R, Sato K, Kuraku S | 2020 | Whale shark blood cell transcriptome | https://www.ncbi.nlm. nih.gov/bioproject/? term=PRJDB8472 | NCBI BioProject, PRJDB8472 |
| Tan M, Read TD, Dove ADM | 2017 | Contig-level assembly | https://www.ncbi.nlm. nih.gov/assembly/GCA_ 001642345.2 | GenBank, LVEK00000000.2 |
| Tan M, Read TD, Dove ADM | 2021 | Scaffold-level assembly | https://www.ncbi.nlm. nih.gov/nuccore/ LVEK00000000.3 | NCBI Nucleotide , GCA_001642345.3 |

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

## Appendix 1

### Genome content

#### Transposable element content

The whale shark genome is relatively large compared to most (but not all) other fish genomes that have been sequenced. The larger genome size relative to *Callorhinchus* was likely driven in part by increases in content of repetitive elements such as transposable elements, as typical in eukaryotes (*Gregory, 2005*). Repetitive elements (*Supplementary file 20*) were annotated using the repeat library construction scripts (Source Code 5). These are slightly modified from those used in MAKER-P (*Campbell et al., 2014*; *Jiang et al., 2016*). This pipeline includes using MITE-Hunter (version 11–2011) to identify MITEs (*Han and Wessler, 2010*), LTRharvest/LTRdigest (both installed through genometools version 1.5.8) to identify 99% similar and 85% similar LTRs (*Ellinghaus et al., 2008*; *Steinbiss et al., 2009*), and RepeatModeler v4.05 to identify other putative repeats (*Smit and Hubley, 2008*). To further classify repeats not initially identified by RepeatModeler, we used RepeatClassifier (which comes with RepeatModeler), and then searched the remaining unclassified repeats against the Dfam database, November 7, 2016, release downloaded February 27, 2017 (*Hubley et al., 2016*) using nhmmer (HMMER v3.1b2) with default settings (*Hubley et al., 2016*). We annotated that the proportion of repetitive elements to the genome length was 50.34% of the genome assembly (*Supplementary file 21*), only slightly higher than previous estimates: *Hara et al., 2018* annotated the proportion of repetitive elements as 46.63% of the genome, and *Weber et al., 2020* annotated the proportion as 49.55% of the genome.

Based on annotation of repetitive elements, we found a much larger proportion of the whale shark genome (>50%) consists of transposable elements compared to *Callorhinchus* genome (28%) (*Supplementary file 21*). This is similar to the proportion found in zebrafish and higher than in human, which have genomes of roughly the same size (*Braasch et al., 2016*; *Howe et al., 2013*). This is also similar to the proportion of repetitive elements previously reported (*Hara et al., 2018*; *Weber et al., 2020*). By contrast, the assembly-free approach based on *k*-mers by GenomeScope estimated only ~789 Mbp repeat length (28%), which is likely an underestimate. Given the draft nature of our assembly, we may still underestimate the proportion of the genome comprised of repetitive elements (*Chalopin et al., 2015*). A large proportion of repetitive elements was also found in the white shark genome (*Marra et al., 2019*).

Though the relationship between overall proportion of transposable elements and the whale shark genome size is consistent with patterns found in other vertebrates, proportions of transposable element classes differed (*Supplementary file 21*). The most well-represented class of repeats of DNA transposons, LINE, SINE, and LTR in the whale shark genome were LINEs (33.25% of the genome), with the most well-represented superfamily of repeats being the CR1 LINEs (21.78% of the genome). CR1 LINEs comprise almost all the transposable element content in the whale shark genome (84.22% of genome content covered by transposable elements), which is high compared to the other vertebrate genomes considered. Compared to *Callorhinchus*, the whale shark has a greater proportion of LINEs (33.25% vs. 12.6%), which is primarily a difference in the proportion of CR1 repeats (21.78% vs. 4.0%) (*Venkatesh et al., 2014*), which has previously been noted to be in high proportion in the whale shark genome (*Marra et al., 2019*; *Weber et al., 2020*). In addition, *Callorhinchus* has a much higher proportion of SINEs than whale shark (13.1% vs. 1.55%) (*Venkatesh et al., 2014*). The whale shark also has a higher proportion of LINEs and CR1s relative to the white shark (33.25% vs. 29.84%; 21.78% vs. 18.75%). The previous assembly of the whale shark had a lower proportion of LINEs and CR1s than the white shark genome (*Marra et al., 2019*), and the increase is likely in part due to the use of long reads in the new assembly. The whale shark also has a relatively high proportion of unclassified repeats among vertebrates (10.8%, most vertebrates <5%) (*Chalopin et al., 2015*). Previous research also identified the large proportion of the genome comprising CR1 LINEs, particularly in introns (*Weber et al., 2020*).

The whale shark genome contains many of the transposable element superfamilies that are widespread in vertebrates, including the DNA transposons TcMariner, hAT, PIF-Harbinger, and Helitron, and many RNA transposons including the LTRs Gypsy, Copia, endogenous retroviruses, and the LINEs Penelope, RTE, CR1, and LINE2 (*Supplementary file 20*). We did not detect some transposable elements found in *Callorhinchus* including the DNA transposons PiggyBac, Sola, and Crypton

and the RNA transposons Dong, and R2. By contrast, we found potential sequences of a number of transposons not found in *Callorhinchus*, including the DNA transposon Novosib and RNA transposons Ngaro, Rex-Babar, and Jockey (*Chalopin et al., 2015*; *Shao et al., 2018*). The presence of Novosib in whale shark extends the presence of this transposon in vertebrates to cartilaginous fishes, as it was previously only detected in teleosts. The whale shark genome appears to contain no full LINE1 repeats, which is consistent with this ancient repeat family being only present among Chondrichthyes as small fragments (*Ivancevic et al., 2016*), however both whale shark and *Callorhinchus* possess Tx1 L1-like repeats (*Chalopin et al., 2015*). Overall, the presences and absences of transposable elements in the whale shark is consistent with the patchy distribution of transposable element superfamilies among vertebrates (*Chalopin et al., 2015*).

## Gene completeness assessment using RNA-seq data

We also assessed mapping of RNA-seq Illumina read data generated from another whale shark individual from blood that were previously generated (*Hara et al., 2018*; *Supplementary file 22*). Sequences were first trimmed using Trimmomatic using the following options: TruSeq3-PE-2. fa:2:30:10 HEADCROP:13 SLIDINGWINDOW:4:20 MINLEN:50. We assembled the blood transcriptome reads by mapping to our genome assembly using HISAT v2.0.5 (*Kim et al., 2015*). The genome was indexed using the command hisat2-build. All trimmed reads (paired reads and unpaired reads after trimming) were aligned to the genome using the command hisat2 with the `-dta` flag to output alignments in a format for StringTie, and alignments were sorted using samtools sort. Sorted HISAT alignments were passed to StringTie v1.3.2b under default settings to assemble transcripts (*Pertea et al., 2015*). We used the gffcompare utility packaged with StringTie to match the StringTie reference-based assembly to the RefSeq annotation and quantify the number of exact matches. We assembled 10,873 transcripts (representing 9606 genes) that exactly matched a RefSeq transcript's internal exon-intron boundaries, a similar number to the 10,990 transcripts (representing 8938 genes) correctly assembled for human whole B cells in blood using the same software (*Pertea et al., 2015*). This is despite the transcriptome being fairly incomplete relative to the 2586 BUSCO v2 conserved vertebrate_odb9 genes (likely because it derives from a single tissue type), with only 1443 (1199 single-copy, 244 duplicated) genes recovered as complete, 309 recovered as fragments, and 1271 orthologs that were missing. This is consistent with core gene analysis (BUSCO, CVG), which suggests that the gene completeness of the whale shark is relatively high.

## Identifying potential missing gene annotations in whale shark and *Callorhinchus*

Through preliminary orthology determination of chordate proteomes using OrthoFinder v2.2.6 (see Materials and methods), we found that numerous gene families conserved in vertebrates were absent in either or both whale shark and *Callorhinchus* RefSeq annotations. Some of these gene families were expected to be in these genomes a priori based on the conserved presence of orthologs in other vertebrates, including some that are known to be present in elasmobranchs based on previous research. In this preliminary orthology determination (not presented), 857 gene families were inferred to be lost in the MRCA of Chondrichthyes, 299 gene families were inferred to be gained in the MRCA of Osteichthyes, 1057 gene families were lost specifically in *Callorhinchus*, and 757 gene families were lost specifically in whale shark. Hence, 1913 of these gene families were absent from the whale shark genome while 2213 were absent from the *Callorhinchus* genome, with an overlap of 1156 gene families that were missing in both. The absence of these gene families in either or both chondrichthyan lineage would affect the inference of the origin and loss of gene families in the MRCA of gnathostomes, the MRCA of Chondrichthyes, and species-specific gains and losses in whale shark and *Callorhinchus*.

To identify putative members of these gene families in whale shark and *Callorhinchus*, we aligned orthologous sequences to both genomes. For whale shark, we aligned *Callorhinchus* proteins for the 757 gene families that would be inferred to be lost in whale shark, and for *Callorhinchus*, we aligned the 1057 whale shark sequences for gene families that were inferred lost in *Callorhinchus*. We also aligned human, coelacanth, gar, and mouse protein sequences for the 1156 gene families that were inferred missing in both chondrichthyan genomes to both *Callorhinchus* and whale shark genomes.

We used genBlast v1.39 to identify putative homologous protein sequences (*She et al., 2011*; *She et al., 2009*), which is a pipeline that utilizes TBLASTN (BLAST+ v2.6.0) to identify putative homologous sequences, then processes these alignments to identify putative orthologues and identify splicing sites. We aligned the orthologous protein sequences to each genome and conservatively selected only the top-ranking putative protein annotated by genBlast for each mapped sequence as a putative ortholog. To exclude proteins annotated by genBlast that were already annotated by RefSeq, we used gffcompare to determine the overlap among annotated regions for genes annotated by genBlast and RefSeq annotations, and we filtered down to only the genBlast annotated proteins that had no overlap with RefSeq annotations (class code of overlap between putative protein and reference sequences coded 'u'). Finally, because there may be multiple sequences annotated for each locus (particularly for the bony vertebrate genes where multiple sequences representing the same gene were aligned to each genome), we filtered putative proteins down to the longest sequence per locus, resulting in 764 sequences annotated in whale shark and 376 sequences in *Callorhinchus*. Genes annotated by GenBlast in whale shark and *Callorhinchus* are provided in *Supplementary file 15* and *Supplementary file 16*, respectively. We then performed orthology assignment again including these additional annotations.

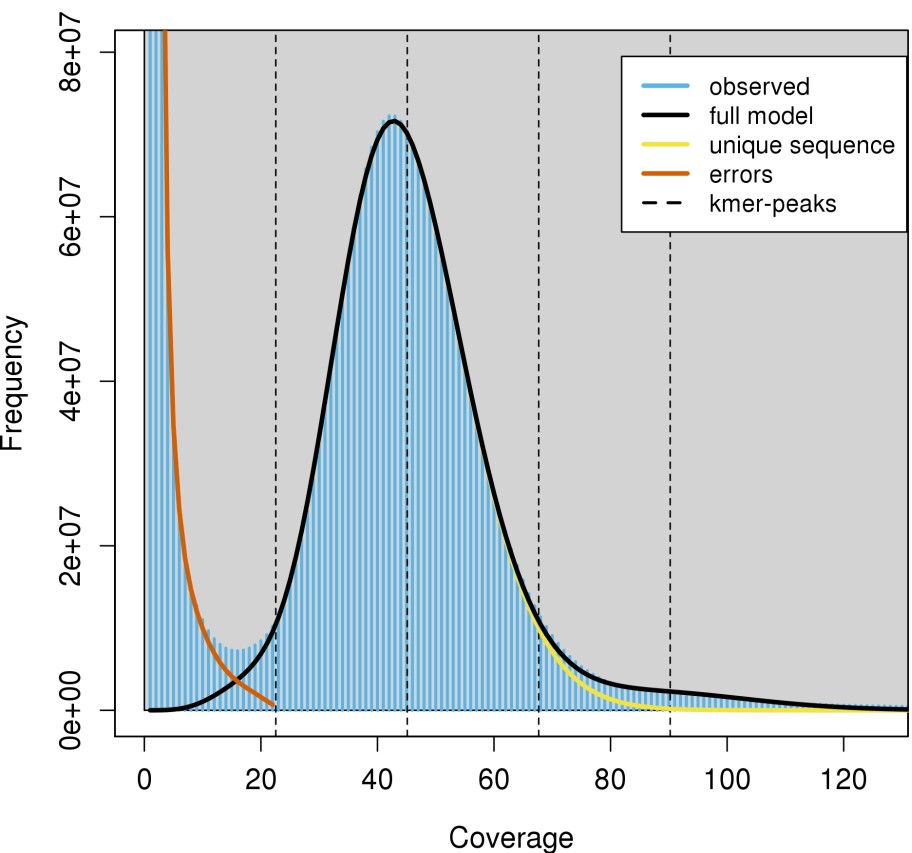

**Appendix 1—figure 1.** Characteristics of the whale shark genome assembly by *k*-mer profiling of raw Illumina reads by GenomeScope. GenomeScope fits a model to estimate genome parameters including heterozygosity (het), an estimated genome size (len), the unique proportion of the genome (uniq; as opposed to the remainder which would be repetitive genome length). Profiling of *k*-mers reveals high coverage sequencing as well as low heterozygosity. Consistent with low heterozygosity, most of the *k*-mers form one peak centered around roughly 40× coverage, and do not form another peak centered at roughly half the coverage that would represent *k*-mers arising from heterozygous alleles.

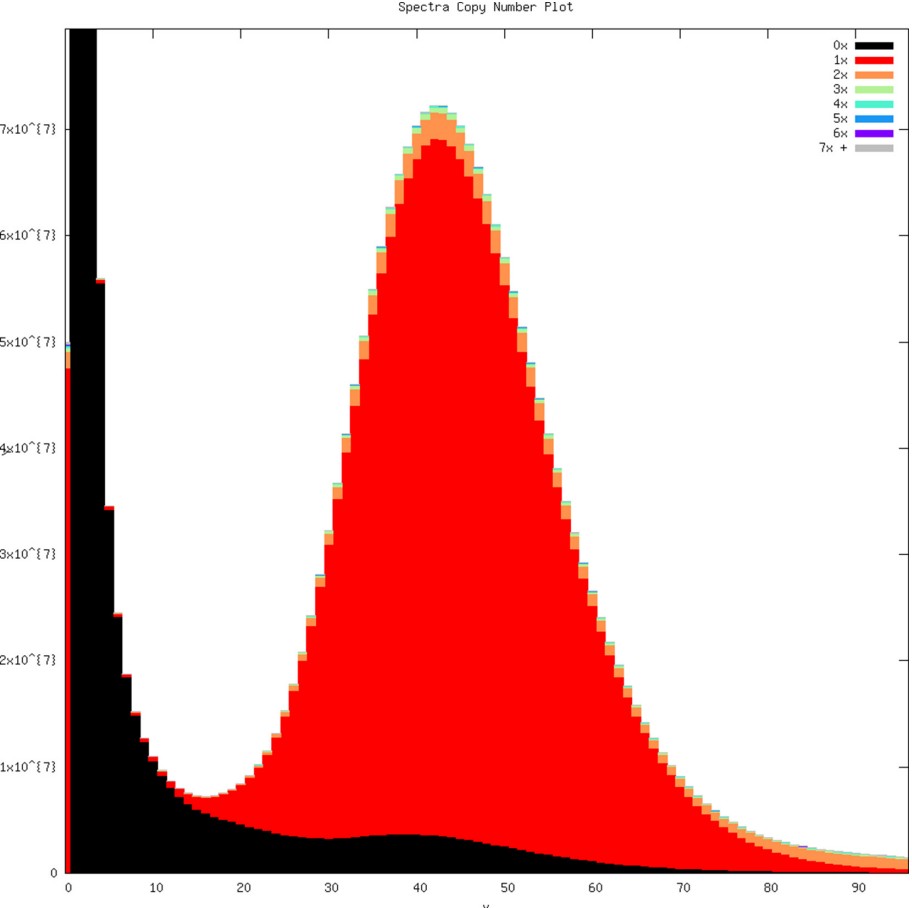

**Appendix 1—figure 2.** *k*-mer profile overlaid with copy number representation within the genome assembly as produced by KAT. *k*-mers arising from error in Illumina raw reads on the left part of the plot are not within the assembly (represented 0×). Most of the *k*-mers in the genome assembly are represented by a single copy (1×, red), suggesting an accurate haploid genome assembly with few diploid alleles assembled as separate contigs.

## Appendix 2

### Comments on functional enrichment of gene families specific to cartilaginous fishes

In contrast to the enrichment of functions for gene families specific to bony vertebrates (gene families gained in bony vertebrates or ancestral genes lost in cartilaginous fishes), we find much less enrichment of terms for gene families specific to cartilaginous fishes. First, we found no enrichment for function or domain terms in the 276 chondrichthyan-derived genes. While it is possible that novel cartilaginous fish genes are not enriched for any particular function, it is also possible that the lack of enrichment may be partially explained by the difference in the number and proportion of genes with annotations: 142 of the 276 (51.4%) gene families that were novel to cartilaginous fishes were annotated, while 301 of the 414 (72.7%) gene families novel to bony vertebrates were annotated, indicating that the novel gene families in cartilaginous fishes were less likely to possess known protein domains or functions. This may have potentially been an effect of the greater divergence of chondrichthyans from well-characterized osteichthyans and the lower level of genomic study that chondrichthyans have received relative to osteichthyans. Therefore, the functional enrichment of gene families that osteichthyan-derived genes may be misled if there is a bias of genes of certain functions to be more difficult to annotate in cartilaginous fishes using InterProScan. This reinforces that some conclusions about the origin and evolution of immune genes in gnathostomes may require more focused study (*Dijkstra, 2014*; *Redmond et al., 2018*).

For the 208 gene families lost in osteichthyans and retained in chondricthyans, only one term was enriched: dynein complex. Given the gene families possessing this are not found in the relatively well-studied bony vertebrate genes and were annotated as genes with unknown names, these orthogroups may also be poorly known. Further genomic study and increased taxon sampling in cartilaginous fishes, chordates, and jawless fishes will help to provide further evidence of the distribution of these putative gene families across vertebrates lost in bony vertebrates.

## Appendix 3

### Additional results and discussion of innate immune pathogen receptors in whale shark

#### NLRs

Human NLRs are intracellular receptors for a wide array of PAMPs and DAMPs (*Caruso et al., 2014*; *Fritz and Kufer, 2015*; *Keestra-Gounder and Tsolis, 2017*; *Proell et al., 2008*; *Ting et al., 2008*). NLRs play key roles in innate immune defense, mainly through regulation of the inflammatory response through NF-κB and inflammasome (an intracellular multiprotein complex that activates inflammatory responses and pyroptosis [inflammation-induced programmed cell death]) (*Caruso et al., 2014*; *Latz et al., 2013*; *Proell et al., 2008*). NLRs are defined by the shared presence of a NACHT domain (although most also possess C-terminal leucine-rich repeats) and are grouped into three families: the NODs, NLRPs (NALPs), and IPAF. Studies of NLR evolution have revealed large-scale lineage-specific expansion of NLRs in some invertebrate lineages (*Huang et al., 2008*; *Rast et al., 2006*; *Yuen et al., 2014*), as well as in teleost fishes (*Howe et al., 2016*; *Laing et al., 2008*). The whale shark genomic data were applied here to assess the cartilaginous fish, and ancestral jawed vertebrate, NLR repertoire.

#### Additional notes on triplicated NOD1

Most strikingly among the NLR results, the whale shark genome harbors three NOD1 genes (UFBOOT=100; *Figure 2—figure supplement 1*). All three of the whale shark NOD1 sequences contain detectable NACHT domains and occupy unique genomic locations. This expansion may permit broader bacterial recognition or provide more nuanced responses to different pathogens. Our detailed analyses including elephant shark, for which we found two NOD1s, indicate that the three NOD1s in whale shark all encode detectable NACHT domains and originated in the ancestor of cartilaginous fishes (*Figure 2—figure supplement 1*). We named the three whale shark genes NOD1-A, -B, and -C. To further confirm the orthology of whale shark NOD1s, we employed synteny analysis which revealed that all three are located on short contigs. NOD1-A and NOD1-B are located on a single contig in the whale shark genome, and are the only genes annotated on this contig (*Figure 2—figure supplement 1*). NOD1-C is located on a different two-gene contig, next to a Cystathionine gamma-lyase (CTH)-like gene (*Figure 2—figure supplement 1*). Our analyses of human and zebrafish NOD1 regions do not support linkage with Cystathionine gamma-lyase (*Figure 2—figure supplement 2*). However, we find that NOD1s in elephant shark are flanked by ZNRF2 and MTURN on one side and this is also the case for human and zebrafish (*Figure 2—figure supplement 2*). Interestingly, whale shark ZNRF2 and MTURN are located at the start of a contig (*Figure 2—figure supplement 2*), so it is possible that they also link to at least one of the NOD1 contigs. Finally, although NOD1-A and -B are located on different contigs than NOD1-C in whale shark, NOD1-A and NOD1-C orthologs in elephant shark (where NOD1-B appears to be lost) are located next to each other (*Figure 2—figure supplement 2*), implying that the three cartilaginous fish NOD1s emerged via tandem duplication events. Human NOD1 plays an important role in intracellular detection of bacterial peptidoglycan among a variety of other agonists. As such, we hypothesized that the three copies of NOD1 in whale shark may permit broader ligand recognition or provide more nuanced response to different pathogens/commensals. To this end, we sought to identify key amino acid changes that might impact function. Sequence motifs previously reported as essential to the function of NOD1 are highly conserved in all three whale shark molecules, including the Walker A and B motifs in the NACHT domain necessary for nucleotide binding and hydrolysis, several LxxLL motifs thought to mediate protein-protein interaction, and residues crucial for binding to the RIP2 adaptor protein and downstream signaling (*Boyle et al., 2013*). Further, the pattern of residue conservation in the leucine-rich repeat (LRR) domains indicates that the whale shark NOD1 molecules, like those of other species, bind ligand on their concave surfaces. However, our attention was drawn to one residue in the LRR domain (E816 in human NOD1) which differed across the whale shark NOD1 molecules. Previous studies have shown that this residue contributes to the preferential binding of different peptidoglycan fragments by mouse and human NOD1 (*Girardin et al., 2005*), supporting our suggestion that the whale shark duplicates have different recognition specificities. Thus,

we hypothesize that all three whale shark NOD1 molecules are functional and share an ancestral mechanism of action but have different recognition specificities, potentiating broader and/or more nuanced responses to intracellular pathogens than species with a single NOD1 gene. Finally, examination of our orthogroups found that the three NOD1 orthologs are also conserved in white shark, brownbanded bamboo shark, and clouded catshark (OG0001251, *Supplementary file 4*).

## NRLP-related expansion in the whale shark

As mentioned above, only a single NLRP-like sequence (for which a NACHT domain was not detected) was identified in whale shark, despite the fact that NLRPs are vital for inflammasome activation in studied species (*Schroder and Tschopp, 2010*). It therefore seems reasonable to suggest that the expanded repertoire of NLRP-related genes in whale shark provides the necessary 'NLRP' inflammasome activators, especially given that we also found a lack of one-to-one orthologs between human NLRP inflammasome activators and zebrafish NLRPs (UFBOOT≥99; *Figure 2—figure supplement 1*). Further, lineage-specific clades of NLRPs are indicative of either concerted or rapid birth-death evolution, suggesting that inflammasome evolution (in the form of gene turnover) is highly influenced by environment (i.e. lineage/life history/environment-specific immune challenges) (*Nei and Rooney, 2005*; *Thomas, 2005*). Such processes have also been observed for other immune genes, for example, antiviral interferons (*Redmond et al., 2019*) and various adaptive immune genes (*Nei et al., 1997*).

## RLRs, DICER, and MAVS

### MDA5 in whale shark

Although we generally found one copy of RIG-1 and LGP2, we found two MDA5-like sequences in our analysis. Additional analyses do not provide sufficient support to indicate the presence of an additional MDA5 gene in whale shark. The two protein sequences do not overlap when aligned to other MDA5/RLRs and appear to be at the start and end of different scaffolds, implying that they may in fact be a single gene separated by a genome assembly gap. No sequence from the transcriptome assembly could bridge the gap between these sequences either, but given that no evidence of additional RLRs in vertebrates has been previously reported (even after another round of whole-genome duplication in teleosts *Jaillon et al., 2004*), it seems most likely that whale shark possesses a single MDA5 gene.

### MAVS in whale shark

Reciprocal BLAST searches between whale shark and the NCBI non-redundant protein database did not reveal a putative MAVS protein in whale shark. Upon more relaxed investigation not requiring reciprocal BLAST hit, one blast hit, from searches of whale shark with MAVS from other vertebrates, revealed a sequence with a CARD domain that appeared similar to those of other MAVS proteins, as well as the CARD domains of MDA5 and RIG-1. Inclusion of this sequence in a phylogenetic analysis of RLR and MAVS CARD domains (using caspase CARDs as outgroups, following *Korithoski et al., 2015*) verified that this was in fact whale shark MAVS (UFBOOT=100; *Figure 2—figure supplement 3*), and further verified the assignment of whale shark MDA5 and RIG-1 (UFBOOT=100 in all cases; *Figure 2—figure supplement 3*). The phylogenetic analysis also placed a coelacanth sequence within the MAVS clade (UFBOOT=100; *Figure 2—figure supplement 3*), despite the previous difficulty in identifying such a sequence (*Boudinot et al., 2014*), suggesting that MAVS is probably ubiquitous in jawed vertebrates. *Callorhinchus* orthologs of all three RLRs, MAVS, and DICER were also identified and/or verified.

## TLRs

TLRs are probably the best known of all innate immune genes, and their functions and evolutionary history have been studied extensively, particularly in comparison to other PRRs (*Leulier and Lemaitre, 2008*; *Roach et al., 2005*; *Vidya et al., 2018*; *Wang et al., 2016*). TLRs are (typically) membrane spanning receptors that recognize disparate but specific conserved structures, for example, TLR3 recognizes viral dsRNA, while TLR4 recognizes bacterial LPS, and TLR9 unmethylated CpG

dinucleotides (*Akira and Takeda, 2004*; *Barton and Medzhitov, 2002*; *Vidya et al., 2018*). Vertebrate TLRs consist of a TIR domain involved in signal transduction and LRRs that permit target recognition (*Akira and Takeda, 2004*; *Vidya et al., 2018*; *Wang et al., 2016*). Evolutionary studies to date suggest that the vertebrate TLR repertoire is highly conserved, with only small changes between species, whereas large-scale lineage-specific expansions have been observed in invertebrates (*Boudinot et al., 2014*; *Huang et al., 2008*; *Rast et al., 2006*; *Roach et al., 2005*; *Wang et al., 2016*). However, a few differences are observed between the teleost and mammal TLR repertoires (partially due to whole-genome duplication in teleosts). The spotted gar genome, a close relative of teleosts that did not share the teleost-specific genome duplication, possesses a mosaic of mammalian and teleost-like TLRs, while an expansion of TLRs in codfishes correlates with loss of CD4 and MHC class II in this lineage (*Boudinot et al., 2014*; *Braasch et al., 2016*; *Malmstrøm et al., 2016*; *Roach et al., 2005*; *Star et al., 2011*; *Wang et al., 2016*). TLRs have been predicted in the *Callorhinchus* genome but await orthology assignment (*Venkatesh et al., 2014*). Previously, we found a TLR similar to TLR13 and TLR21 in our previous whale shark genome draft assembly (*Read et al., 2017*); comparison of this with the sequences included in the vertebrate TLR tree below indicates that this previously studied sequence corresponds to TLR21. The TLR repertoire of whale shark was assessed here to better understand the evolution of vertebrate TLRs and contextualize this with respect to invertebrate TLR expansions and the emergence of adaptive immunity.

## TLRs in *Callorhinchus* vs. whale shark

In the TLR tree presented here, *Callorhinchus* does not possess orthologs of TLR21 or TLR29, but does possess orthologs of TLR14/18 and TLR25 (*Boudinot et al., 2014*; *Wcisel et al., 2017*). Thus, while the TLR repertoires of whale shark and elephant shark are quite well conserved, repertoire differences do exist; understanding the impact of which requires functional characterization of TLR14/18, TLR25, and TLR29. Extrapolating from our data, the MRCA of whale shark and elephant shark must have possessed at least 12 TLRs from different ancestral jawed vertebrate lineages. This implies that cartilaginous fishes possess a highly similar number of TLRs to mammals (~10). However the whale shark TLR repertoire is novel in that it contains a mix of orthologs to the classical mammalian TLRs and 'fish-specific' TLRs (similar to spotted gar *Braasch et al., 2016*; *Wcisel et al., 2017*), as well as to TLR27 and the new TLR29. These last two appear to be absent from both mammals and teleosts, but present in the so-called 'living fossil' lineages (e.g. sharks, coelacanths, gars).

## TLR9

Intriguingly, despite TLR9 evolution being well studied, in our analysis we observe two maximally supported TLR9 sister clades (*Figure 2—figure supplement 4*), each of which contains an array of mutually exclusive vertebrate taxa except both contain spotted gar orthologs (*Wcisel et al., 2017*). We also found evidence that a third TLR9 group falls sister to these (UFBOOT=94; *Figure 2—figure supplement 4*). For the newly discovered jawed vertebrate TLR9 lineages, we suggest the following nomenclature: TLR9a (includes the originally characterized mammalian TLR9), TLR9b (or TLR30; includes the new whale shark TLR9), TLR9c (or TLR31) (*Figure 2—figure supplement 4*). A similar scenario appears to have played out in TLR13 evolution, where three potential lineages are also detectable TLR13a (including mammalian TLR13), TLR13b (TLR32; including reptilian TLR13), and TLR13c (TLR33) (*Figure 2—figure supplement 4*), although only a single lineage appears to have existed in the jawed vertebrate ancestor.

## Ancestral reconstruction of TLR repertoires

Based on our rooted TLR tree, it is possible to estimate a minimal ancestral jawed vertebrate TLR repertoire, by inferring gene loss based on the species possessing a sister gene (*Peterson and Sperling, 2007*; *Redmond et al., 2017*). As an example, in the TLR tree presented here, TLR7 and TLR8 are sister clades, with both containing representatives of each of the major jawed vertebrate groups (*Figure 2—figure supplement 4*). As such although cartilaginous fish orthologs of TLR7 and TLR8 exist (*Figure 2—figure supplement 4*), if the tree contained no TLR8 orthologs, their existence could be inferred, because the presence of a cartilaginous fish TLR7 ortholog means that these genes split prior to speciation between cartilaginous fishes and bony vertebrates. Using this

phylogenetic logic, it is apparent that at least 17 TLRs existed in the last common ancestor of jawed vertebrates, TLR1/6/10, TLR2/28, TLR3, TLR4, TLR5/5SL (possibly both), TLR7, TLR8, TLR9 (possibly three of these), TLR11/12/16/19/20/26, TLR13, TLR14/18, TLR15, TLR21, TLR22/23 (possibly both), TLR25, TLR27, TLR29 (inferred from *Figure 2—figure supplement 4*). This number is much higher than what is commonly found in most jawed vertebrate lineages (excluding gar, teleosts, and particularly Atlantic cod, for the reasons stated above *Boudinot et al., 2014*; *Braasch et al., 2016*; *Roach et al., 2005*; *Star et al., 2011*; *Wcisel et al., 2017*). This result thus supports a scenario of convergently decreasing TLR repertoire complexity in the major jawed vertebrate lineages since their last common ancestor.

There are some caveats to this count, for example, root placement is taken to be reasonably accurate, invertebrates were not included, and common vertebrate gene tree errors such as teleosts grouping sister to all other jawed vertebrates, or slowly evolving species grouping together were permitted. Additional potential errors may also exist, such as the placement of TLR15, which here is taken to be lost or not yet found in all non-reptile (incl. avian) lineages, but could conceivably be a highly divergent ortholog of another TLR group that is not easily phylogenetically placed due to extreme rate asymmetry (*Holland et al., 2017*; *Manousaki et al., 2011*; *Redmond et al., 2018*).

In addition to considering the ancestral jawed vertebrate TLR repertoire, the inclusion of lamprey TLRs in the tree meant that an estimate of the minimal ancestral vertebrate TLR repertoire could also be considered (*Figure 2—figure supplement 4*). Using the same approach as above, and considering the same caveats, the last common ancestor of all vertebrates likely had at least 15 TLRs, spanning the full array of TLR superfamilies, and presumably their functions; TLR1/6/10, TLR2/28, TLR3, TLR4, TLR5/5SL, TLR7/8, TLR9, TLR11/12/16/19/20/22/23/26, TLR13, TLR14/18, TLR15, TLR21/29, TLR24, TLR25, TLR27 (*Figure 2—figure supplement 4*). This result implies an increase in TLR repertoire between the last common ancestors of all vertebrates and jawed vertebrates. Lineage-specific duplications in jawed vertebrates, for example, TLR7 and TLR8, TLR21, and TLR29 (each of which are co-orthologs of lamprey lineages for which we suggest the names TLR7/8-like [or TLR34], and TLR21/29-like [or TLR35]; *Figure 2—figure supplement 4*), have played an important role in this. However, the uniqueness of the lamprey genome (*Smith et al., 2013*) as well as the relatively poor sampling of jawless vertebrates could also contribute to this small discrepancy.

## Adaptive immunity and the evolution of PRR repertoires

Early hypotheses to explain the stark differences between the relatively conserved PRR repertoires in vertebrates and the remarkable PRR expansions in invertebrates (particularly in deuterostomes) suggested that the adaptive immune system negated the need for diversification of novel PRRs in vertebrates. Here, although we found evidence for a core set of PRRs that appear to have been 'locked in' during early vertebrate evolution alongside the appearance of the new adaptive immune system, we found that NLR expansions in jawed vertebrates are common and extensive, while jawed vertebrate RLRs (although major expansion of this family in invertebrates also seems not to have occurred) and TLRs appear to fit a model where expansions are constrained. These differences may be due to the complexity of interactions between adaptive and innate immune systems, as well as those between host and pathogen.

For example, the RLRs are the most conserved set of examined PRRs in our analyses, being nearly identical in repertoire in all jawed vertebrates. Experimental MDA5 duplication improves the immune response, but also accelerates autoimmunity (*Crampton et al., 2012*). MDA5 is capable of functionally replacing RIG-1 following loss, implying a certain level of redundancy (*Xu et al., 2016*). Together, this suggests that the RLR repertoire in vertebrates is constrained to maintain balance between a maximally robust response while avoiding autoimmunity. However, it is important to note that large expansion of this family in invertebrates has not been reported, and so it is possible that adaptive immunity has had an inconsequential role in orchestrating RLR repertoire evolution.

TLR evolution appears to occur more readily through gene loss than expansion in vertebrates, perhaps fitting a scenario where TLRs are lost if they become obsolete or are subverted by a pathogen. In general, TLRs fit well with the idea of adaptive immunity replacing or constraining the need for innate innovation through extensive gene duplication, however new TLRs have emerged independently of genome duplication during vertebrate evolution (e.g. TLR1/6/10 duplications, TLR11/

12 duplication), suggesting that despite the presence of adaptive immunity there is still a place for new germline-encoded TLRs.

NLR expansions have occurred multiple times during jawed vertebrate evolution and to independent NLR family members, firmly rejecting a situation where adaptive immunity relieves the need for PRR expansions. These expansions (assuming rapid turnover) may be required to maintain functional relevance for rapid detection of (new) lineage-specific pathogens or inflammasome activation (*Latz et al., 2013*; *Schroder and Tschopp, 2010*).

## Appendix 4

### Body size evolution in Chondrichthyes

Gigantism in vertebrates including elephants and whales is associated with a shift to a new rate regime of body size evolution (*Puttick and Thomas, 2015*; *Slater et al., 2017*). We tested if whale sharks are also unexpectedly large given background rates of body size evolution in cartilaginous fishes, which would imply that other aspects of its biological evolution may have also shifted in association with this gigantism.

Patterns of body size evolution were estimated using a published distribution of 500 time trees and log-transformed body mass data for chondrichthyans with 10 fossil calibrations (*Stein et al., 2018*). Analysis across a posterior distribution of trees allows for accounting for uncertainty in phylogenetic inference in estimating patterns of body size evolution. Body mass data for chondrichthyans from a previous study (*Stein et al., 2018*) were kindly provided by Chris Mull. Mass data were log-transformed for comparative analyses. Time-varying rates of body size evolution were estimated using BAMM 2.5.0 and BAMMtools 2.1.6 (*Rabosky, 2014*; *Rabosky et al., 2014*). For each of the trees, we used setBAMMpriors to calculate appropriate prior probabilities that were scaled for each tree. Each MCMC run was run with four chains for at least 30M generations, sampling once every 10K generations, and discarding the first 10% of samples as burn-in. We assessed convergence using the effectiveSize() function in coda v0.19–1 and determined if any parameters were below 200 effective samples. Runs with insufficient samples were re-run at 50M generations. All runs reached convergence (i.e. had over 200 effective samples) with 50M generations. After discarding burn-in, 2000 samples were extracted from the event data for each tree. We determined the marginal odds ratio of shifts for all branches, including the branch leading to the whale shark, to assess the support for a rate shift in body size evolution leading to the whale shark. To compute mean rates of body size evolution across branches for each posterior sample, we estimated the rate of evolution including the single branch representing the whale shark using getCladeRates(). To compute the mean rates of body size evolution in the background rate of body size evolution, for each posterior sample, we computed the rates of body size evolution across all branches that shared the background rate regime (the rate regime that originated at the root). Scripts to perform analysis perform are provided (*Source code 6*).

Numerous independent shifts in body size evolution were inferred in the chondrichthyans, including along the branch leading to the whale shark. We recovered a significant shift in the rate of body size evolution along the branch leading to the whale shark with a mean marginal odds ratio of 241.65, demonstrating strong support for a shift in the rate of body size evolution in the branch leading to the whale shark and demonstrating that the gigantism in the whale shark is not inferred to be the result of neutral, background rates of body size evolution. The mean rate of body size evolution along the whale shark branch across all posterior samples and all 500 tree samples (1M total samples) was 0.532 log-grams per million years, relative to 0.117 log-grams per million years estimated for the background rate in chondrichthyans (*Appendix 4—figure 1*). The difference in mean rate of body size evolution between the whale shark and the background across all samples is 4.31 times the background rate. These results support that the gigantic body size of the whale shark is due to a shift to a novel rate regime of body size evolution.

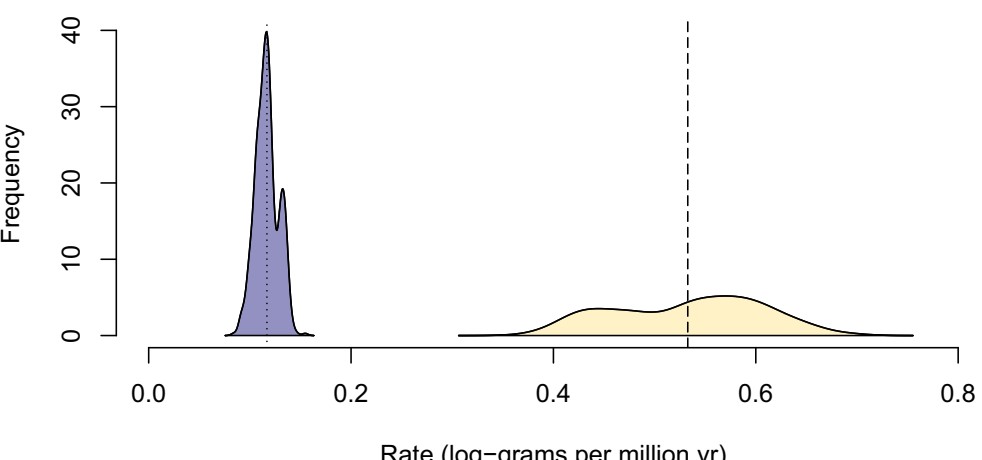

**Appendix 4—figure 1.** Distribution of mean estimated rate of body size evolution of the posterior distribution estimated for each tree sample (n = 500) for the background in Chondrichthyes (green) and for the whale shark (blue). Dotted line indicates mean estimated rate for Chondrichthyes across all tree samples, while the dashed line indicates mean estimated rate for the whale shark.

