## [Decision Letter]

**Acceptance summary:**

This manuscript is of interest in the field of comparative genomics as it provides novel genomic resources for the whale shark, which belongs to a group of vertebrates with a unique biology, which to date had limited available genomic data. This work provides important insights into the evolution of several key vertebrate-specific gene families, including genes involved in pathogen recognition receptors (PRRs) and cancer suppression.

**Decision letter after peer review:**

Thank you for submitting your article "The whale shark genome reveals patterns of vertebrate gene family evolution" for consideration by *eLife*. Your article has been reviewed by 3 peer reviewers, one of whom is a member of our Board of Reviewing Editors, and the evaluation has been overseen by Patricia Wittkopp as the Senior Editor. The following individual involved in review of your submission has agreed to reveal their identity: Michael Hiller (Reviewer #2).

Essential revisions:

1. Despite the sequencing of long-range reads, the authors did not scaffold the contig assembly. What is the reason for this? Did the authors use the mate pair data from previous studies (e.g. Read et al)?

2. Cancer gene families seem to be prone to shifts anywhere (line 493). Therefore, the authors should test if swapping the large animals for their normal sized sister taxa (e.g. whale shark for bamboo shark, minke whale for dolphin) also results in a significant enrichment in cancer genes. If that is the case, then there is probably no or only a weak connection between cancer gene family evolution and body size.

3. Expand the explanation regarding the evolutionary analysis of PRRs (reviewer #3, point1).

4. Expand the methods part, providing detailed information for reproducibility and transparency reasons (see reviewer #3, point 2). At the same time, integrate more of the methods section into the Results section, providing essential information about the methods and databases used.

5. Invertebrate deuterostomes are often used in the paper as an example of animals that have an expanded PRR repertoire due to the lack of an adaptive immune system. However, despite often using invertebrate deuterostomes as an outgroup, the authors never quantify PRR repertoire shifts between invertebrate deuterostomes and the vertebrate groups used for comparison. Quantifying these shifts between groups will strengthen such arguments. Also, a more effective outgroup might be non-jawed vertebrates lacking a lymphocite-based adaptive immune system provided there is enough genomic data to interrogate PRR diversification.

*Reviewer #1 (Recommendations for the authors):*

The world-wide increase of genomic surveys across vertebrates (see genome 10K) have significantly raised the bar for genomic studies and publication of individual genomes. Long-read technologies should be also coupled with HiC and provide chromosome-level genome assemblies. In this respect, this manuscript does not reach the standards that would grant a publication in a journal like *eLife*. Furthermore, the present whale shark genome is an upgrade from a previous assembly, hence does not meet the criterion of "novel species". Additionally, the work presented does not provide significant insights into the biology of the species studies.

*Reviewer #2 (Recommendations for the authors):*

1) The authors compared gene families with shifts in large animals (and possibly other lineages) and show enrichments for cancer genes. This is interesting, however cancer gene families seem to be prone to shifts anywhere (line 493). Therefore, the authors should test if swapping the large animals for their normal sized sister taxa (e.g. whale shark for bamboo shark, minke whale for dolphin) also results in a significant enrichment in cancer genes. If that is the case, then there is probably no or only a weak connection between cancer gene family evolution and body size.

*Reviewer #3 (Recommendations for the authors):*

1. The evolutionary analysis of PRRs is not sufficiently explained. For instance, the authors say that they used subsampling of a previously used set for the TLR analysis (line 699). The nature of the original set is not described satisfactorily (i.e. species, database, etc; see point 4 below), the method and depth of subsampling is not explained (i.e. software used, % of coverage, random or guided), and the scientific rationale for such a choice instead of using a full set is not given. In addition, there are other potential issues emerging, notably why does the TLR analysis include more species than for NLR and RIG-like? When trying to infer NLR repertoire shifts in vertebrates, the authors compare humans (mammal, bony vertebrate), zebrafish (bony fish), and whale shark (cartilaginous fish). However, comparing only three groups seems insufficient to make conclusions regarding the evolution of NLR repertoire shifts. Why not include the breadth of species used in the TLR phylogenetic analysis? The selection of other included species in that analysis needs to: (i) be transparent, including information about how the other species/sequences are selected and obtained and (ii) be unified, i.e. include the same taxonomic depth for all considered groups.

2. There are a number of missing pieces of information that need to be provided for reproducibility and robustness. The method section, particularly, needs to be completely revamped and developed.

a. Please make sure all version numbers of used software's are explicitly stated in the methods section. Although many are specified, version numbers are not available for a number of used software or packages, and version numbers (or date of access) also need to be provided for annotation databases (e.g. genBLAST, BLAST, DIAMOND, biomaRt, gVolante, etc.).

b. Please indicate all necessary information for which species were included, which genome builds were used for the other species used for phylogenetic analyses in the main methods. Please also explain why genome builds for Refseq, Ensembl 89 and 99 are mixed (see Table S3).

c. Please provide additional details/information on the refseq pipeline they used to annotate the genome (Gene prediction section in methods), including software version, parameters, etc. Intriguingly, although the gene prediction section in methods mentions the refseq annotation pipeline, Table S2 mentions another annotation software, MAKER (not mentioned anywhere in the methods section). Please make sure to describe accurately which methods were used.

d. "We also performed assembly-free estimation of genome size, heterozygosity, and repeat content.", line 586. This is not enough detail for a methods section to ensure reproducible analysis. Please provide an explanation of how this was done, using which softwares, etc.

e. In table S1, please indicate DNA source abd raw read data etc without referring to previous paper. Indicate all here for proper comparison, or the table is not providing the needed information.

f. Finally, it is key for long-term reproducibility that all the code for this study be made available to readers (e.g. as a github repository or a supplemental zip file).

3. Too much of core analyses are supplied as supplementary notes, partially redundant to the main text and partially containing information that clearly belongs in the main text. The results need to be integrated and discussed in the main paper, otherwise they will be mostly lost to the readers.

4. The authors make the claim that, "While we found evidence for a core set of NLRs in jawed vertebrate, our analyses also suggest that the immunologically relevant NLR repertoire shifts may be as common in vertebrates as they are in deuterostome invertebrates, despite the presence of the adaptive immune system." Invertebrate deuterostomes are often used in the paper as an example of animals that have an expanded PRR repertoire due to the lack of an adaptive immune system. However, despite often using invertebrate deuterostomes as an outgroup, the authors never quantify PRR repertoire shifts between invertebrate deuterostomes and the vertebrate groups used for comparison. Quantifying these shifts between groups will strengthen such arguments. Also, a more effective outgroup might be non-jawed vertebrates lacking an adaptive immune system provided there is enough genomic data to interrogate PRR diversification.

---

## [Author Response]

Essential revisions:1. Despite the sequencing of long-range reads, the authors did not scaffold the contig assembly. What is the reason for this? Did the authors use the mate pair data from previous studies (e.g. Read et al)?

Initial scaffolding attempts did not lead to significant improvements and so were not reported. We performed a new scaffolding using Platanus that led to a much larger improvement, and report these statistics in the paper (line 137), added the relevant scaffold information to Table S1, and added corresponding detail to the methods (lines 645-648).

2. Cancer gene families seem to be prone to shifts anywhere (line 493). Therefore, the authors should test if swapping the large animals for their normal sized sister taxa (e.g. whale shark for bamboo shark, minke whale for dolphin) also results in a significant enrichment in cancer genes. If that is the case, then there is probably no or only a weak connection between cancer gene family evolution and body size.

We thank the reviewers for this suggestion and have performed this analysis. It turns out that swapping the normal-sized sister taxa in this test finds no significant enrichment of cancer genes, which further supports the result in the paper that the gene families that had shifted along branches leading to giant taxa were enriched for cancer genes and that there was a role of body size (odds ratio = 0.88, p = 0.624; lines 543-547). We also took it one step further by drawing the same number of branches as giant branches at random for 100 permutation to determine a null distribution of odds ratios of cancer gene enrichment. We found that the observed odds ratio of 2.66 was greater than 98% of odds ratios from random permutation (in other words, p = 0.02; lines 547-552). Hence, the finding of many gene families related to cancer gene function shifting along giant taxa seems to be a finding not expected by chance. The methods corresponding to these new analyses are within the Methods section lines 947-951.

3. Expand the explanation regarding the evolutionary analysis of PRRs (reviewer #3, point1).

See below for specific comments addressing this concern.

4. Expand the methods part, providing detailed information for reproducibility and transparency reasons (see reviewer #3, point 2). At the same time, integrate more of the methods section into the Results section, providing essential information about the methods and databases used.

See reviewer #3 point 2 for specific responses. The request to integrate more of the methods into the results was also improved throughout the results, including indicating GenomeScope was used for k-mer based heterozygosity estimation (lines 126, 141), freebayes used for SNP calling (line 148), orthogroup determination using OrthoFinder from primarily Ensembl and RefSeq databases (which are used in most downstream analyses; line 175), orthogroup annotation using InterProScan and Kinfin (lines 227-228), the two-cluster test using LINTRE (line 459), PAML to estimate differences in rates of amino acid substitution (line 482), and café to estimate gene family size evolution (line 498).

5. Invertebrate deuterostomes are often used in the paper as an example of animals that have an expanded PRR repertoire due to the lack of an adaptive immune system. However, despite often using invertebrate deuterostomes as an outgroup, the authors never quantify PRR repertoire shifts between invertebrate deuterostomes and the vertebrate groups used for comparison. Quantifying these shifts between groups will strengthen such arguments. Also, a more effective outgroup might be non-jawed vertebrates lacking a lymphocite-based adaptive immune system provided there is enough genomic data to interrogate PRR diversification.

See below for specific comments addressing these concerns.

Reviewer #1 (Recommendations for the authors):The world-wide increase of genomic surveys across vertebrates (see genome 10K) have significantly raised the bar for genomic studies and publication of individual genomes. Long-read technologies should be also coupled with HiC and provide chromosome-level genome assemblies. In this respect, this manuscript does not reach the standards that would grant a publication in a journal like eLife.

We recognize this limitation, however the application of long-range technologies (e.g. HiC) are outside the scope of the present work. It should be noted that the present assembly is a significant improvement over the previous assembly. Further, although not chromosome-level, this genome assembly has already (this study), and will continue to, significantly advance our understanding of the relatively understudied cartilaginous fish lineage.

Reviewer #2 (Recommendations for the authors):1) The authors compared gene families with shifts in large animals (and possibly other lineages) and show enrichments for cancer genes. This is interesting, however cancer gene families seem to be prone to shifts anywhere (line 493). Therefore, the authors should test if swapping the large animals for their normal sized sister taxa (e.g. whale shark for bamboo shark, minke whale for dolphin) also results in a significant enrichment in cancer genes. If that is the case, then there is probably no or only a weak connection between cancer gene family evolution and body size.

See response to Essential Revision 2

Reviewer #3 (Recommendations for the authors):1. The evolutionary analysis of PRRs is not sufficiently explained. For instance, the authors say that they used subsampling of a previously used set for the TLR analysis (line 699). The nature of the original set is not described satisfactorily (i.e. species, database, etc; see point 4 below), the method and depth of subsampling is not explained (i.e. software used, % of coverage, random or guided), and the scientific rationale for such a choice instead of using a full set is not given. In addition, there are other potential issues emerging, notably why does the TLR analysis include more species than for NLR and RIG-like? When trying to infer NLR repertoire shifts in vertebrates, the authors compare humans (mammal, bony vertebrate), zebrafish (bony fish), and whale shark (cartilaginous fish). However, comparing only three groups seems insufficient to make conclusions regarding the evolution of NLR repertoire shifts. Why not include the breadth of species used in the TLR phylogenetic analysis? The selection of other included species in that analysis needs to: (i) be transparent, including information about how the other species/sequences are selected and obtained and (ii) be unified, i.e. include the same taxonomic depth for all considered groups.

We thank the reviewer for highlighting this oversight and have updated the text to make the subsampling performed more transparent. Subsampling of the extensive dataset presented by Wang et al., (2016) was performed to reduce the computational burden of our analysis while retaining a dataset representative of both TLR diversity and major vertebrate lineages. To do this we manually removed species from closely related groups that shared identical or highly similar TLR repertoires. We also excluded species for which only very few TLRs have been described but where all such TLRs overlap with a very closely related species for which a more complete TLR repertoire was known. We have added a statement detailing this to the materials and methods section (lines 784-808). Further, we have added supplementary table (Suppementary File 9) showing the species we chose to include and exclude from the original Wang et al., dataset, information on the number of TLRs present for each species, and information on the lineages to which each species belong. We hope this will allow readers to better understand our choices.

Regarding sampling of the different PRR types, our TLR analysis included more species than the other PRRs simply because TLR repertoires are far better characterized across vertebrates. The density of species included in RLR analyses was informed by the study of Mukherjee et al., (2014) and was designed to incorporate key representatives of vertebrate and invertebrate lineages. We agree that a broader species sample for NLRs would have been nice, however several factors made expansion of our dataset difficult. First, NLR repertoires are relatively poorly described across vertebrates and their identification is confounded by the fact that the NLR-defining NACHT domain cannot always be reliably detected – even some well characterized human NLRs do not have easily detectable NACHT domains. Indeed, this was also the case for some of the putative whale shark NLRs we identified (although our evidence for NLR expansions are independent of any uncertainty arising from this). This difficulty is particularly important considering that the number of NLRs in any given vertebrate genome can vary greatly and can be very large, e.g., while the human genome harbors just over 20 NLR genes, zebrafish have well over 400. Thus, determining NLR repertoires for even a small number of species is a considerable undertaking and puts such analyses beyond the scope of the present study. However, even though our study includes only three species, this is enough to determine that lineage-specific expansions have occurred in vertebrates (despite the presence of the adaptive immune system) even if the finer timeline on which these events has occurred is not yet clear. A statement to this effect has been added to the materials and methods section.

Please also see our response to point #4 below for further discussion around factors which need to be considered in future.

2. There are a number of missing pieces of information that need to be provided for reproducibility and robustness. The method section, particularly, needs to be completely revamped and developed.a. Please make sure all version numbers of used software's are explicitly stated in the methods section. Although many are specified, version numbers are not available for a number of used software or packages, and version numbers (or date of access) also need to be provided for annotation databases (e.g. genBLAST, BLAST, DIAMOND, biomaRt, gVolante, etc.).

We have provided version numbers or data of access for all software, packages, and

databases.

b. Please indicate all necessary information for which species were included, which genome builds were used for the other species used for phylogenetic analyses in the main methods. Please also explain why genome builds for Refseq, Ensembl 89 and 99 are mixed (see Table S3).

Supplementary table 4 lists all species used in phylogenomic analyses. RefSeq sequences are specifically included because these taxa were not available on Ensembl at the time the analyses were conducted. Analyses were initially performed primarily from genomes on Ensembl 89 as well as additional genomes that were not available on Ensembl and were hence included from RefSeq. Comments from peers after posting the initial preprint suggested the inclusion of other taxa; at this point, the available version of Ensembl was Ensembl 99.

c. Please provide additional details/information on the refseq pipeline they used to annotate the genome (Gene prediction section in methods), including software version, parameters, etc. Intriguingly, although the gene prediction section in methods mentions the refseq annotation pipeline, Table S2 mentions another annotation software, MAKER (not mentioned anywhere in the methods section). Please make sure to describe accurately which methods were used.

We clarified in the methods that RefSeq provided these annotations to us to make explicit that we did not run the annotation pipeline, but rather that RefSeq ran the annotation pipeline and provided the results to us. We added the version (7.3) to the methods. RefSeq did not report the parameters they used in performing the annotation for us, but we added a citation that they recommended that documents their pipeline. We requested additional information from RefSeq, and they gave the following response:

"Unfortunately, the RefSeq annotation pipeline is not a portable piece of software (we have plans to make is so). It involves a complex suite of software, and is highly tied to core NCBI components and databases. Although I understand the reviewer’s request, the annotation steps can’t correctly be reproduced by an outside user. Given that your manuscript is more about the interpretation of the data that is public rather that the annotation process itself, I hope that this will be a satisfactory to the reviewer."

The MAKER repeat annotation pipeline was specifically used to annotate repeats and Supplementary table 2 is specifically only cited in the supplement. Relevant information on the MAKER annotation pipeline to the supplement is on lines 10-19, a citation for step-by-step instructions for running the pipeline by Jiang et al., 2019 is cited (line 12), and the commands used to perform the annotation are now provided (Supplementary File 10). To clarify, the MAKER repeat annotation pipeline does not use MAKER, but was called that by authors presumably as they used it to provide repeats for annotation in MAKER downstream.

d. "We also performed assembly-free estimation of genome size, heterozygosity, and repeat content.", line 586. This is not enough detail for a methods section to ensure reproducible analysis. Please provide an explanation of how this was done, using which softwares, etc.

We moved the information from the supplementary text to the main text (lines 658-666).

e. In table S1, please indicate DNA source abd raw read data etc without referring to previous paper. Indicate all here for proper comparison, or the table is not providing the needed information.

We duplicated the information from column 2 for Read et al., 2017 to the other columns to make it unambiguous we used the same sample.

f. Finally, it is key for long-term reproducibility that all the code for this study be made available to readers (e.g. as a github repository or a supplemental zip file).

Scripts or configure files are provided for repeat annotation, gene gain and loss, gene gain and loss enrichment, treePL, PAML, CAFE analyses in supplementary file 10. Flags or text description of options used are provided in the methods for straightforward program runs, including read trimming (lines 642-644), read alignment (lines 647-648), genblast (line 695-696), sequence alignment (lines 826-829), and maximum likelihood phylogenetic analyses (lines 838-845, 871-872).

3. Too much of core analyses are supplied as supplementary notes, partially redundant to the main text and partially containing information that clearly belongs in the main text. The results need to be integrated and discussed in the main paper, otherwise they will be mostly lost to the readers.

Much of this was done to attempt to keep the length of the main text shorter. We reduced the supplementary information by moving information to the main text or excluding some repetitive information. This includes information about genome assembly statistics (previously Appendix 1A), assessing gene completeness (previously Appendix 1C), and much of the repetitive text regarding gene family evolution in vertebrates (much of Appendix 2). We retain supplementary results in the Appendix relating to repetitive element content, some aspects of gene family evolution, some notes on immune gene evolution that provide additional context, and the rates of body size analysis. These are all indicated specifically where the Appendix is cited in the text. We entirely removed some sentences from the main text or supplement that are not necessary to describe to interpret the main results.

4. The authors make the claim that, "While we found evidence for a core set of NLRs in jawed vertebrate, our analyses also suggest that the immunologically relevant NLR repertoire shifts may be as common in vertebrates as they are in deuterostome invertebrates, despite the presence of the adaptive immune system." Invertebrate deuterostomes are often used in the paper as an example of animals that have an expanded PRR repertoire due to the lack of an adaptive immune system. However, despite often using invertebrate deuterostomes as an outgroup, the authors never quantify PRR repertoire shifts between invertebrate deuterostomes and the vertebrate groups used for comparison. Quantifying these shifts between groups will strengthen such arguments. Also, a more effective outgroup might be non-jawed vertebrates lacking an adaptive immune system provided there is enough genomic data to interrogate PRR diversification.

We agree that an analysis quantifying the tempo of PRR repertoire evolution to identify where rate shifts occur between jawed vertebrates, jawless vertebrates, and deuterostome invertebrates would be very interesting. However, tackling this analysis would a huge undertaking and, we feel, a study in its own right. Such a study would require densely and relatively evenly sampled PRR repertoires from across bilaterian and vertebrate phylogeny and time calibration of accurate gene trees to reliably infer gene duplication times, as well as consideration of species level shifts in gene duplication and loss rates (e.g. vertebrate genome evolution is considered to be generally slower). Doing so adequately would rely on the availability of new high-quality genomes and characterization of PRR repertoires for many species (a difficult task, as discussed in the response to point #1), and as such falls beyond the scope of the present manuscript. To better acknowledge this, we have softened statements such as that quoted by the reviewer to avoid implying that the prevalence and timing of rate shifts is known. Our primary aim was to determine if PRR expansions are a feature of vertebrate PRR evolution and our analyses clearly show this is the case.

We have also better discussed the need for additional work and how our inferences are guided by current knowledge of deuterostome invertebrates, including through addition of a new table to the main text with PRR counts from the whale shark, other jawed vertebrates, jawless vertebrates (whose use as an outgroup is complicated because they possess an alternate adaptive immune system and possibly a distinct genome duplication history from jawed vertebrates – but as noted in our new table, they have a generally similar PRR repertoire to humans), and invertebrates, so that readers can see the raw repertoire numbers in key species that have been previously studied.